# Active Learning of Convex Halfspaces on Graphs

**Maximilian Thiessen**
Research Unit of Machine Learning
TU Wien, Vienna, Austria
`maximilian.thiessen@tuwien.ac.at`

**Thomas Gärtner**
Research Unit of Machine Learning
TU Wien, Vienna, Austria
`thomas.gaertner@tuwien.ac.at`

## Abstract

We systematically study the query complexity of learning geodesically convex halfspaces on graphs. Geodesic convexity is a natural generalisation of Euclidean convexity and allows the definition of convex sets and halfspaces on graphs. We prove an upper bound on the query complexity which is linear in the treewidth and the minimum hull set size but only logarithmic in the diameter. We show tight lower bounds along well-established separation axioms and identify the Radon number as a central parameter of the query complexity and the VC dimension. While previous bounds typically depend on the cut size of the labelling, all parameters in our bounds can be computed from the unlabelled graph. We provide evidence that ground-truth communities in real-world graphs are often convex and empirically compare our proposed approach with other active learning algorithms.

## 1 Introduction

We systematically characterise active learning of geodesically convex halfspaces on graphs. While unlabelled graphs such as social networks are readily accessible, obtaining many labels is tedious and labour intensive. Active learning reduces the labelling effort by iteratively selecting informative data points to be labelled. Graph-based active learning, in particular, has been successfully applied on communication [Regol et al., 2020] and protein-protein-interaction networks [Vazquez et al., 2003, Xiong et al., 2014]. While most previous approaches assume that the number of edges with differently labelled vertices, the *cut size*, is small [Afshani et al., 2007, Guillory and Bilmes, 2009, Dasarathy et al., 2015], we take a different approach and instead assume *geodesic convexity*: For every pair of vertices with the same label it holds that every vertex on every shortest path between them also has that label. This assumption is used, for example, by biologists on gene similarity networks [Zhou et al., 2002] and cancer-related protein-protein-interaction networks [Li et al., 2012, 2013]. It also typically holds for connected subgraphs of collaboration networks [Marc and Šubelj, 2018, Šubelj et al., 2019] and our preliminary experiments confirmed that many communities in community detection datasets are convex. We derive bounds on the number of queries required to determine all labels, that is, the *query complexity*, using concepts from *convexity theory* and *geodesic convexity spaces*, which naturally generalise the regular Euclidean convexity. While learning convex sets and halfspaces in Euclidean spaces has been intensively investigated in the machine learning community for over half a century, geodesic convexity on graphs is understudied. For a concise presentation of our theoretical results in this paper, we concentrate on active learning of *halfspaces* on graphs, that is, the vertex set is partitioned into two classes which are both geodesically convex.

After introducing the concepts of graph and abstract convexity theory in Section 2, we derive general label-independent upper and lower bounds on the query complexity. We discuss that such general bounds can be loose on specific graphs and derive tighter lower bounds along the *separation axioms* of abstract convexity theory in Section 3. This allows us to substantially reduce the gap between our upper and lower bounds. Note that Euclidean spaces cannot be structured in a similar way: while the separation axioms partition the space of all graphs into non-empty sets, all separation axioms

35th Conference on Neural Information Processing Systems (NeurIPS 2021).

hold in all real vector spaces including the Euclidean spaces. We discuss related topics like previous active learning bounds that depend on the cut size between the classes, active learning of halfspaces in Euclidean spaces, non-active learning of geodesically convex halfspaces, and other related learning theoretic concepts in Section 4. Notably, we provide a bound on the VC dimension of geodesically convex halfspaces given by the *Radon number*. After presenting an empirical validation of our assumptions and approach in Section 5, we conclude with a discussion of future work.

## 2   Convexity spaces

In this section, we introduce the required concepts of convexity theory. For a more thorough introduction on convexity theory we refer the reader to van de Vel [1993] and Pelayo [2013].

For a set $X$ and a family $\mathcal{C} \subseteq 2^X$ of subsets, the pair $(X, \mathcal{C})$ is a *convexity space* if *(i)* $\emptyset, X \in \mathcal{C}$, *(ii)* $\mathcal{C}$ is closed under intersection, and *(iii)* $\mathcal{C}$ is closed under unions of sets totally ordered by inclusion. For finite set systems, property *(iii)* always holds. Any set in $\mathcal{C}$ is called *convex*. If a set $C$ and its complement $X \setminus C$ are convex, both are called *halfspaces*. Two disjoint sets $A, B$ are *halfspace separable* if there exists a halfspace $C$ such that $A \subseteq C$ and $B \subseteq X \setminus C$. A mapping $\sigma : 2^X \to 2^X$ is a *convex hull* (or *closure*) *operator* if for all $A, B \subseteq X$ with $A \subseteq B$ *(i)* $\sigma(\emptyset) = \emptyset$, *(ii)* $\sigma(A) \subseteq \sigma(B)$, *(iii)* $A \subseteq \sigma(A)$, and *(iv)* $\sigma(\sigma(A)) = \sigma(A)$. Any convexity space $(X, \mathcal{C})$ induces a convex hull operator by $\sigma(A) = \bigcap \{C \mid A \subseteq C \in \mathcal{C}\}$. A set $A \subseteq X$ is convex, that is $A \in \mathcal{C}$, if and only if is equal to its convex hull, $A = \sigma(A)$.

A set $H \subseteq X$ is a *hull set* if its convex hull is the whole space, $\sigma(H) = X$. For $A, B \subseteq X$, the set $A/B = \{x \in X \mid A \cap \sigma(B \cup \{x\}) \neq \emptyset\}$ is the *extension of $A$ away from $B$*. For $a, b \in X$, the extension $\{a\}/\{b\}$ is also called a *ray $a/b$*. Two disjoint sets $A_1, A_2$ form a partition of $A \subseteq X$ if $A_1 \cup A_2 = A$. The partition $A_1, A_2$ of $A$ is a *Radon partition* if $\sigma(A_1) \cap \sigma(A_2) \neq \emptyset$. The *Radon number* is the minimum number $r$ such that any subset of $X$ of size $r$ has a Radon partition.

A particular type of convexity is an *interval convexity*. Apart from the convex hull $\sigma(\cdot)$ it has an *interval mapping* $I : X \times X \to 2^X$ such that for all $x, y \in X$, *(i)* $x, y \in I(x, y)$ and *(ii)* $I(x, y) = I(y, x)$. $I(x, y)$ is the *interval* between $x$ and $y$. We denote $I(A) = \bigcup_{a,b \in A} I(a, b)$. A set $C$ in an interval convexity space is convex if and only if $C = I(C)$. The convex hull is given by $\sigma(A) = \bigcup_{k=1}^{\infty} I^k(A)$, where $I^1(\cdot) = I(\cdot)$ and $I^{k+1}(\cdot) = I(I^k(\cdot))$, A well-known instance of interval convexity spaces are *metric spaces* $(X, d)$. There, the interval contains all the points for which the triangle inequality holds with equality: $I_d(x, y) = \{z \in X \mid d(x, y) = d(x, z) + d(z, y)\}$. In Euclidean space this corresponds to all points on a line segment and leads to the classical notion of convex sets.

We study metric spaces induced by graphs. The *geodesic convexity* (or shortest path convexity) of a connected graph $G = (V, E)$ is given by the interval mapping $I_d$, where $d : V^2 \to \mathbb{R}$ is the *shortest path distance* in $G$. Let $x, y \in V$. For unweighted graphs $d(x, y)$ is the minimum number of edges on any $x$-$y$-path and for graphs with edge weights, $w : E \to \mathbb{R}_{>0}$, it is the minimum total edge weight of any $x$-$y$-path. A set of vertices $C \subseteq V$ is, thus, (geodesically) convex if and only if $C$ contains every vertex on every shortest path joining vertices in $C$, corresponding again to the Euclidean case.

We denote by $V(G)$ the vertex set of a graph $G$, the Radon number of the induced geodesic convexity space of a graph as $r(G)$, and the size of the minimum hull set as $h(G)$. Rays $a/b$ in graphs can be defined using the shortest path distance $d$: In unweighted graphs $a/b = \{v \in V \mid d(a, v) < d(b, v)\}$ and in weighted graphs $a/b = \{v \in V \mid d(b, v) = d(b, a) + d(a, v)\}$. A vertex $v$ is *extreme* if $\{v\}$ forms a halfspace, that is, $V \setminus \{v\}$ is convex. We denote the set of extreme vertices of $G$ as $\text{Ext}(G)$. In unweighted graphs, $v$ is extreme if and only if its neighbours form a clique. The *diameter* $d(G)$ of a weighted or unweighted graph $G$ is the maximum number of edges in any shortest path in $G$.

## 3   Active learning halfspaces on graphs

Having introduced the necessary concepts from convexity theory, we now present the main theoretical results of the paper. In *active node classification*, a graph $G = (V, E)$ with unknown vertex labels $\lambda : V \to \{0, 1\}$ is given and the goal is to accurately predict all labels using as few as possible vertex queries. Edges can be weighted or unweighted. The learner queries the vertices one by one: it iteratively selects vertices $v \in V$ and receives their label $\lambda(v)$. In this paper, we consider active learning on an undirected connected graph where the vertices with same label form geodesically

convex sets. The classes are, thus, halfspaces and we call the labelling *halfspace separable*. Our results in this section are upper and lower bounds on the number of queries required to deduce all labels $\lambda$ of the graph. We denote the number of queries required to identify any halfspace separable labelling in the worst-case, the *query complexity*, by $\mathrm{qc}(G)$. We derive two general upper bounds (Proposition 1, Theorem 5) and one lower bound (Theorem 7) on $\mathrm{qc}(G)$ that hold for any halfspace separable labelling. To reduce the gap between the upper and lower bounds (Proposition 8), we derive increasingly tighter lower bounds along separation axioms (Theorem 10). In this section, we consider only undirected and connected graphs. Full proofs and a discussion of the multiclass, directed, and disconnected setting are in the supplementary material.

## 3.1   Upper bounds on the query complexity

To derive a simple upper bound, we note that one immediate consequence of the halfspace assumption is that any shortest path $P$ can have at most one *cut edge*, that is, an edge with differently labelled endpoints. To deduce all labels of $P$, we first query its endpoints. If they have the same label, we know that all vertices on $P$ have this label. Otherwise, we find the cut edge by binary search with at most $\lceil \log d(G) \rceil$ queries, as the length of $P$ is at most the diameter $|V(P)| - 1 \leq d(G)$. Here, $\log$ is the base 2 logarithm. We can generalise this approach to the whole graph using *shortest path covers* [Thiessen and Gärtner, 2020], which is a set $\mathcal{S}$ of shortest paths whose vertices cover the graph: $\bigcup_{P \in \mathcal{S}} V(P) = V(G)$. Performing binary search on each path in $\mathcal{S}$ gives our first upper bound.

**Proposition 1.** *For any weighted graph $G$ with minimum shortest path cover $\mathcal{S}^*$ the query complexity can be bounded as $\mathrm{qc}(G) \leq |\mathcal{S}^*|(2 + \lceil \log d(G) \rceil)$.*

By considering a complete graph with edge weights such that the position of the cut edges on all shortest path in the minimum shortest path cover can be chosen independently of each other, we can show that this bound is tight:

**Proposition 2.** *For any $\ell, s \in \mathbb{N}$, there exists a weighted graph $G$ with diameter $d(G) = \ell$ and minimum shortest path cover $\mathcal{S}^*$ of size $s$ such that $\mathrm{qc}(G) \geq |\mathcal{S}^*| \log d(G)$.*

However, the graph in Proposition 2 is an artificial worst-case example. For most graphs, we can do much better as the labels of vertices on different paths in the cover can typically not be chosen independently of each other. Consider a cubic hypergrid graph $G$, that is, the Cartesian product of $k$ paths with $\ell$ vertices each. More than $|V(G)|/d(G) + 1 \geq \ell^k/k\ell$ shortest paths are needed to cover $G$ but $2 + \lceil \log(k\ell) \rceil$ queries suffice to identify all labels, as fixing one cut edge determines all others.

Our main idea to improve the bound and derive a better algorithm for active learning of geodesically convex halfspaces is to deduce additional labels using convex hulls and extensions after each query. Algorithm 1 realises this idea. It first queries the vertices of a hull set (line 1). If all labels in the hull set are the same, we conclude that all vertices in the graph have the same label (line 2-3). Otherwise, it performs binary search on a shortest path between two vertices with different labels to identify a cut edge $\{a, b\}$ (line 4-5). Finally, Algorithm 1 queries the remaining vertices. It initialises the sets $A$ and $B$ with the convex hull of the rays $a/b$ and $b/a$, respectively, corresponding to vertices with already known labels (line 6). In each iteration of the main loop, the algorithm queries a vertex $v$ in the set $\hat{W}_{=ab} = V(G) \setminus (\sigma(a/b) \cup \sigma(b/a))$, consisting of all vertices whose labels we cannot directly deduce through convex hulls and extensions, and updates the sets $A$ and $B$ using the new vertex (line 7-12). The number of queries spend in the first five lines are summarised in Lemma 3.

**Lemma 3.** *Let $G$ be a weighted graph with halfspace separable labels. Using $h(G) + \lceil \log d(G) \rceil$ queries, we can either find a cut edge or determine that all vertices of the graph have the same label.*

*Proof.* The convex hull of any minimum hull set is the whole graph. If all vertices in a hull set have the same label, then all vertices in the graph have this label. If not, we can take two vertices with different labels and find a cut edge on a shortest path between them with $\lceil \log d(G) \rceil$ queries. $\square$

Lemma 4 shows that the initialisation and updates of $A$ and $B$ in Algorithm 1 are indeed valid.

**Lemma 4.** *Let $G$ be a weighted graph with halfspace separable labels given by a halfspace $C$ and let $A \subseteq C$ and $B \subseteq V(G) \setminus C$. It holds that $\sigma(A/B) \subseteq C$.*

*Proof sketch.* $A/B \subseteq C$ as $A \cap \sigma(B \cup \{x\}) \subseteq C \cap (V(G) \setminus C) = \emptyset$ for all $x \in V(G) \setminus C$. This implies $\sigma(A/B) = \bigcap \{C' \mid A/B \subseteq C', \ C' \text{ convex}\} \subseteq C$ by definition. $\square$

---

**Algorithm 1:** Ray-based Active Halfspace Learning on Graphs

---

**Input:** graph $G$, oracle access to labels $\lambda$
**Output:** halfspaces corresponding to both classes

1   Compute a hull set $H$ and query its vertices
2   **if** $\forall h, h' \in H : \lambda(h) = \lambda(h')$ **then**
3      $\lfloor$   **return** $(\emptyset, V(G))$

4   Choose $h, h' \in H$ such that $\lambda(h) \neq \lambda(h')$
5   Perform binary search queries on any shortest $h$-$h'$-path to find a cut edge $\{a, b\}$
6   $A := \sigma(a/b), \quad B := \sigma(b/a)$
7   **while** $A \cup B \neq V(G)$ **do**
8      query any vertex $v \in V(G) \setminus (A \cup B)$
9      **if** $\lambda(v) = \lambda(a)$ **then**
10        $\lfloor$   $A := \sigma((A \cup \{v\})/B), \quad B := \sigma(B/(A \cup \{v\}))$
11      **else**
12        $\lfloor$   $A := \sigma(A/(B \cup \{v\})), \quad B := \sigma((B \cup \{v\})/A)$

13   **return** $(A, B)$

---

We can bound the remaining number of vertices queried by Algorithm 1 using the following set

$$W^*_{=ab} = \{v \in \hat{W}_{=ab} \mid \nexists w \in \hat{W}_{=ab} \setminus \{v\} \text{ such that } v \in (w/a) \cap (w/b)\}$$

as it is enough to know the labels of the vertices in $W^*_{=ab}$ to deduce all labels of $\hat{W}_{=ab}$. Theorem 5 shows that Algorithm 1 is correct and summarises our main upper bound.

**Theorem 5.** *Let $G$ be a weighted graph. For the query complexity it holds that:*

$$\text{qc}(G) \leq h(G) + \lceil \log d(G) \rceil + \max_{\{a,b\} \in E(G)} |W^*_{=ab}|.$$

*Proof sketch.* By Lemma 3, we can use $h(G) + \lceil \log d(G) \rceil$ queries to find the first cut edge $\{a, b\}$. Every time Algorithm 1 queries a vertex $v \in \hat{W}_{=ab} \setminus W^*_{=ab}$ there is a $w \in W^*_{=ab}$ by definition such that $v \in (w/a) \cap (w/b)$. It holds that $w \notin A \cup B$, as otherwise $v \in (w/a) \cap (w/b) \subseteq A \cup B$, which contradicts line 8 of Algorithm 1, $v \in V(G) \setminus (A \cup B)$. Thus, in each iteration at least one new vertex from $W^*_{=ab}$ will be added to $A$ or $B$. This bounds the number of iterations by $|W^*_{=ab}|$. $\square$

Comparing our two upper bounds, we see that Theorem 5 is preferable to Proposition 1, as long as $|W^*_{=ab}|$ is small. To see this, note that the endpoints of paths in a minimum shortest path cover $\mathcal{S}^*$ form a hull set, implying $h(G) \leq 2|\mathcal{S}^*|$. If $|W^*_{=ab}|$ is small, the bound of Theorem 5 is additive in the dominating terms $h(G) + \lceil \log d(G) \rceil \leq 2|\mathcal{S}^*| + \lceil \log d(G) \rceil$, not multiplicative as in Proposition 1.

To state the bound for the unweighted case in more common parameters, we relate $|W^*_{=ab}|$ to the well-studied *treewidth* $\text{tw}(G)$ [Bodlaender, 1996], which measures the 'tree-likeness' of a graph. It is known, that the largest $k \in \mathbb{N}$, such that the complete bipartite graph $K_{2,k}$ is a minor of a graph $G$, is at most twice the treewidth [Bodlaender et al., 1997]: $k \leq 2\,\text{tw}(G)$. This bounds the size $|W^*_{=ab}| \leq 2\,\text{tw}(G)$ for any edge $\{a, b\}$, as the vertices $\{a, b\} \cup W^*_{=ab}$ form a minor $K_{2,|W^*_{=ab}|}$. Consequently, graphs with small treewidth like molecules [Horváth and Ramon, 2010] and infrastructure-based networks [Maniu et al., 2019] have small $W^*_{=ab}$. Corollary 6 summarises this result.

**Corollary 6.** *Let $G$ be an unweighted graph. For the query complexity it holds that $\text{qc}(G) \leq h(G) + \lceil \log d(G) \rceil + 2\,\text{tw}(G)$.*

### 3.2   Lower bounds under separation axioms

Having discussed our upper bounds on the query complexity, we now turn to a simple lower bound based on the extreme vertices $\text{Ext}(G)$ of the graph, recall each of them is a halfspace by definition.

**Theorem 7.** *For any weighted graph $G$, it holds that $\text{qc}(G) \geq |\text{Ext}(G)|$.*

Considering a graph containing paths that all coincide only in the same two endpoints, we can show that this bound is tight and that the gap between it and our upper bound can be arbitrarily large.

Table 1: Tight bounds for common unweighted graph families.

| graph family | sep. axiom | bound on qc($G$) | remark |
|---|---|---|---|
| trees | $S_4$ | $\Theta(\log d(G) + |\operatorname{Ext}(G)|)$ | $\operatorname{Ext}(G)$ are exactly the leaves |
| $K_{2,3}$ minor-free | $S_4$ | $\Theta(\log d(G) + h(G))$ | including outerplanar graphs |
| partial cubes | $S_3$ | $\Theta(\log d(G) + h(G))$ | $\mathcal{O}(\cdot)$ holds for all bipartite graphs |
| weakly median | $S_4$ | $\Theta(\log d(G) + h(G) + r(G))$ | $\mathcal{O}(\cdot)$ holds for meshed graphs |

**Proposition 8.** *For any $\ell, s, k, e \in \mathbb{N}$, there exists an unweighted graph $G$ with $d(G) \geq \ell$, minimum shortest path cover $\mathcal{S}^*$ of size at least $s$, $h(G) \geq k$, $|\operatorname{Ext}(G)| = e$, and $\operatorname{qc}(G) \leq \max\{2, e\}$.*

This shows that this lower bound is best possible in general, even if $d(G)$, $h(G)$, and $\mathcal{S}^*$ are large. One main insight of our work is that we achieve increasingly tighter lower bounds by structuring the set of all graphs along *separation axioms*, which characterise the ability of a convexity space to separate sets via halfspaces. This follows a related notion in topological spaces.

**Definition 9** (Separation axioms [van de Vel, 1993])**.** *A convexity space $(X, \mathcal{C})$ is:*

$S_1$ *if and only if each singleton $x \in X$ is convex.*

$S_2$ *if and only if each pair of distinct elements $x, y \in X$ is halfspace separable.*

$S_3$ *if and only if each convex set $C$ and elements $x \in X \setminus C$ are halfspace separable.*

$S_4$ *if and only if any two disjoint convex sets are halfspace separable.*

If $S_1$ holds, which is the case for the geodesic graph convexity, the remaining axioms are increasingly stronger, that is, $S_2 \Leftarrow S_3 \Leftarrow S_4$. We call a graph $S_i$, for $i = 1, \ldots, 4$, if the induced geodesic convexity space satisfies the respective separation axiom. While real vector spaces such as the Euclidean space satisfy all four separation axioms [Kakutani, 1937], there are graphs that are $S_i$ but not $S_{i+1}$ for $i = 1, 2, 3$ [Bandelt, 1989], see supplementary material for some examples. Structuring all graphs along these separation axioms gives us increasingly stronger lower bounds.

**Theorem 10.** *For every weighted graph $G$ the following holds for the query complexity $\operatorname{qc}(G)$:*

- *if $G$ is $S_2$, then $\operatorname{qc}(G) \geq \max\{\log d(G), |\operatorname{Ext}(G)|\}$,*

- *if $G$ is $S_3$, then $\operatorname{qc}(G) \geq \max\{\log d(G), h(G)\}$, and*

- *if $G$ is $S_4$, then $\operatorname{qc}(G) \geq \max\{\log d(G), h(G), r(G) - 1\}$.*

*Each bound is tight in the respective family and stronger axioms lead to tighter bounds.*

*Proof sketch.* In $S_2$ graphs, any edge in a fixed but arbitrary shortest path with size of the diameter can be bisected with a halfspace. Thus, $\log d(G)$ queries are required to find the chosen edge. In $S_3$ graphs, any point $h \in H^*$ can be separated from the remainder of a minimum hull set $H^*$, thus the whole hull set $H^*$ must be queried in the worst-case. In $S_4$ graphs, any set that has no Radon partition can be labelled arbitrarily. To see that the bounds are increasingly tighter, notice that all extreme vertices are by definition contained in any hull set and hence $|\operatorname{Ext}(G)| \leq h(G)$. As there are graphs with $h(G) = k$ and $r(G) = t$ for any $k, t \in \mathbb{N}$ with $k, t \geq 4$, both parameter can be dominating. $\square$

Theorems 5 and 10 imply tight bounds (Table 1) for graph families in metric graph theory [Bandelt and Chepoi, 2008], convexity theory [Chepoi, 1994], and machine learning [Seiffarth et al., 2019].

## 3.3 Computational aspects

Above, we derived lower and upper bounds on the query complexity of learning halfspaces on graphs $G = (V, E)$. We now discuss computational aspects of our algorithms and bounds.

**Shortest-path-cover-based bound** To achieve the bound of Proposition 1, we need to compute a minimum shortest path cover of the given graph, which is an open problem [Manuel, 2021]. We show that greedily covering the graph with longest shortest paths gives a logarithmic approximation:

**Theorem 11.** *For a given weighted graph $G$, we can compute an $(1 + \ln(d(G) + 1))$-approximation for the minimum shortest path cover $\mathcal{S}^*$ in time $\mathcal{O}(|V|^4)$.*

Thus, using this approximation we achieve the query complexity bound $\mathrm{qc}(G) \leq |\mathcal{S}^*|(2 + \lceil \log d(G) \rceil)^2$ in polynomial time. To evaluate this bound, we also need to compute the diameter $d(G)$, which can be achieved by a generalised Dijkstra algorithm, see supplementary material.

**Hull-set-based bound** The main computational bottleneck of Algorithm 1 is the computation of a minimum hull set. Computing the minimum hull set size $h(G)$ is APX-hard [Coelho et al., 2015]. For many graph families however, such as *distance-hereditary* [Kanté and Nourine, 2016], *interval*, and *cographs* [Dourado et al., 2009] the problem is solvable in polynomial time. Moreover, the problem can be solved in polynomial time if any of the following parameters is bounded by a constant: the treewidth [Kanté et al., 2019], the vertex cover size, the *neighbourhood diversity* [Araujo et al., 2013b], and the number of induced paths in bounded vertex subsets [Araujo et al., 2013a]. The remaining runtime of Algorithm 1 is dominated by the computation of convex hulls and extensions. A single convex hull computations can be performed in time $\mathcal{O}(|V|^3)$ by first computing the full distance matrix and then evaluating the triangle inequality for each triplet of vertices. Extensions can be computed through $|V|$ convex hull evaluations. Algorithm 1 achieves, thus, the bound of Theorem 5 in polynomial time, as long as we can efficiently compute a minimum hull set. On unweighted graphs we additionally achieve the bound in Corollary 6 efficiently in this case. Alternatively, we can compute any, possibly non-optimal, hull set $H$ in polynomial time to achieve the bound $|H| + \lceil \log d(G) \rceil + \max_{\{a,b\} \in E(G)} |W^*_{=ab}|$.

**Lower bounds** To evaluate our lower bounds in Theorem 10, we also need to compute $|\mathrm{Ext}(G)|$ and $r(G)$. Computing the set of extreme vertices $\mathrm{Ext}(G)$ is possible in time $\mathcal{O}(|V||E|)$ by evaluating the neighbourhood of each vertex. Computing the Radon number of a graph is even hard to approximate within any factor sublinear in $|V|$ [Coelho et al., 2015]. However, Duchet and Meyniel [1983] bounded the Radon number in terms of the size of the largest clique minor, that is, the Hadwiger number, which in turn can be upper bounded by the treewidth.

**Proposition 12.** *For any weighted graph $G$, it holds that $r(G) \leq 2\,\mathrm{tw}(G) + 2$.*

Thus $r(G)$ is computable in polynomial time for bounded treewidth graphs. To decide which lower bound applies for a given graph, we need to check if the graph is $S_i$, for $i = 1, \ldots, 4$. For the cases $i = 1, 2, 3$, it is unclear whether this is possible in polynomial time. Only for $S_4$ an algorithm with runtime $\mathcal{O}(|V(G)|^7)$ is known [Seiffarth et al., 2019]. Just deciding whether a graph has a *proper* halfspace, which is neither empty nor the full graph, is NP-hard [Artigas et al., 2011]. Both problems are efficiently decidable on planar and bipartite graphs [Glantz and Meyerhenke, 2017].

## 4 Discussion

In this section, we discuss our assumptions as well as results related to our learning problem.

**Convexity in real-world datasets** In some applications convexity-based assumptions are already implicitly or explicitly used. For example, it is well-known that shortest paths in gene similarity networks largely preserve functional relationships [Zhou et al., 2002]. Similarly on protein-protein-interaction networks, shortest paths between cancer-related genes are used to identify a candidate set of novel genes, which are likely to be cancer-related, as well [Li et al., 2012, 2013]. Aside from biological networks, Marc and Šubelj [2018] and Šubelj et al. [2019] recently found that connected subgraphs of certain real-world graphs like collaboration networks are often convex. To provide further evidence for convexity in real-world graphs, we performed preliminary experiments on different networks with ground truth communities. Table 2 shows the number of convex communities in six datasets from SNAP [Leskovec and Krevl, 2014]. We found that on the DBLP scientific collaboration dataset more than $85\%$ of the 5000 communities are convex, supporting the results of Šubelj et al. [2019]. On the Amazon product network we have similar results with roughly $80\%$. On the Youtube network we found that roughly $60\%$ of the communities are convex. On the remaining

Table 2: Number of convex communities.

| dataset | convex communities | dataset | convex communities |
|---------|--------------------|---------|--------------------|
| DBLP | 4308/5000 | Amazon | 3999/5000 |
| Youtube | 2990/5000 | LiveJournal | 1649/5000 |
| Orkut | 363/5000 | Eu-core | 7/42 |

three datasets, however, less than a third of the communities are convex. Clearly, convexity on graphs is application dependent and further study with real-world networks is necessary.

**Relationships to Euclidean convexity**  Learning halfspaces in Euclidean space is one of the oldest learning problems initiating algorithms like the perceptron [Rosenblatt, 1958, Novikoff, 1962] and support vector machines [Boser et al., 1992, Cortes and Vapnik, 1995], and is still an active research area [Daniely, 2016, Diakonikolas et al., 2019, Hopkins et al., 2020]. A simple lower bound for the number of queries required to learn the labels of a finite set $X \subseteq \mathbb{R}^m$ labelled accordingly to a halfspace is $\Omega(m \log |X|)$ [Hopkins et al., 2020]. However, for worst-case instances such as points on the circle, already $\Omega(|X|)$ queries are required to identify all labels [Dasgupta, 2005], corresponding to our hull set lower bound. The query complexity can be improved through membership query synthesis [Angluin, 1988], which allows the learner to query additionally any point in $\mathbb{R}^m \setminus X$. Using them, Hopkins et al. [2020] showed that in expectation $\mathcal{O}(m \log^2(m) \log |X|) + m^{1+o(1)}$ queries are enough to infer all labels, essentially matching the lower bound. There is no direct equivalent of membership queries in the transductive graph setting. It is in general not possible to embed a graph into $\mathbb{R}^m$ or the other way around while preserving halfspaces. An example can be seen in Figure 1c and further discussion can be found in the supplementary material. Together with the fact that Euclidean convexity spaces are always $S_4$ while graphs do not have to be, there is no immediate way to transform general bounds on the query complexity from one setting to the other.

**Previous cut-based bounds**  Most previous bounds for active learning on graphs are linear in the number of cut edges $C$ or *cut vertices* $\partial C$, which are the vertices incident to the cut edges. Therefore, these bounds can become vacuous when the cut is large, independent of halfspace separable labels. We will illustrate the bounds on the unweighted $2 \times k$ grid, that is, the Cartesian product of a path with $k$ vertices and a path with 2 vertices. On the one hand, our upper bound of Theorem 5 yields a query complexity of $2 + \log(k + 1)$. On the other hand, the bounds of Afshani et al. [2007] and Dasarathy et al. [2015] result in at least $|\partial C|$ queries even under additional balancedness assumptions on the labels. Thus, the bounds are at least $|V|$ for the halfspace corresponding to the $1 \times k$ grid. The bound of Guillory and Bilmes [2009] is for the non-iterative active setting, where a set $L \subseteq V$ of vertices is queried all at once. Let $\delta(T)$ denote the set of edgs with exactly one endpoint in $T \subseteq V$. They bound the error by $|C|/\Psi(L)$ using the min-cut-based prediction strategy [Blum and Chawla, 2001], where $\Psi(L) = \min_{\emptyset \neq T \subseteq (V \setminus L)} |\delta(T)|/|T|$. As long as $|L| \leq |V|/8$, we will have $\Psi(L) \leq 4/8$ and for the previously mentioned halfspace $|C| = k$. Thus, the error bound will be at least $2k = |V|$. This shows that the three discussed previous bounds can be vacuous and exponentially worse than our achieved bound. All three are particularly sensible to large cut sizes. The sole cut is thus not sufficient to measure the complexity inherent in the labelling.

**Margin-based bounds**  Bressan et al. [2021] study the query complexity of identifying convex clusters in the $\varepsilon$-nearest-neighbour of a semi-metric space $(X, d)$. Even though they rely on additional assumptions, like a large margin, their upper bound is very similar to our main result. However, they only provide lower bounds on the query complexity on specific worst-case instances, while our lower bounds hold in general under the different separation axioms. They use *same-cluster* queries, which are binary queries on a pair of vertices to test whether they belong to the same cluster. Same-cluster queries and regular vertex queries are equivalent in terms of the query complexity up to a multiplicative factor given by the number of classes. To derive a bound they use restrictions beyond our halfspace assumption. In particular, Bressan et al. [2021] assume that the weight of any cut edge is $> \beta\varepsilon$ for a $\beta \in (0, 1]$ and use an interval $I^\gamma(x, y)$ with margin $\gamma \in (0, 1]$, which consists of all vertices that lie on a path at most $(1 + \gamma)$ as long as the shortest $x$-$y$-path, to define convex sets. The regular geodesic convexity corresponds to $\gamma = 0$. They also require a constant number of so-called *seed* queries that take a class label and a set $U \subseteq X$ as input and return a vertex from $U$

with the specified label or certify that $U$ does not contain such a vertex. Under these assumptions they derive an $\mathcal{O}\left(\log|X| + (6/\beta\gamma)^{\text{dens}(X)}\right)$ upper bound on the query complexity, where $\text{dens}(X)$ is the *density dimension* of the space [Gottlieb and Krauthgamer, 2013]. Even though Bressan et al. [2021] consider a more restricted hypothesis space, there exist graph families where our bound in Theorem 5 is exponentially better. For example, on the $2 \times k$ grid, the halfspace corresponding to the $1 \times k$ grid enforces a margin of $\gamma < 2/k$. This results in a bound of at least $\log(2k) + (3k)^{\text{dens}(X)}$ with $\text{dens}(X) \geq 1$ while our bound (Theorem 5) predicts that $2 + \log(k+1)$ queries are enough. Even with a large $\gamma$ our bound remains better on this example.

**Alternatives to hull sets**    As computing minimum hull sets is APX-hard in general [Coelho et al., 2015], we discuss two alternatives to perform the first step in Algorithm 1, that is, finding two vertices with different labels. For that, Afshani et al. [2007] and Dasarathy et al. [2015] assume that the labels are *balanced*, that is, the relative size $\beta$ of the smallest labelled class is close to $1/2$. This assumption is in their case required to find two vertices with different labels query-efficiently with high probability $1 - \alpha$, for $\alpha \in (0, 1)$. We do not require this assumption as we can use any hull set to achieve the same deterministically. However, if $\beta$ is known, we can state an alternative probabilistic upper bound. Instead of spending $h(G)$ queries we can sample a set $L$ of size $\left\lceil \frac{\log(\beta\alpha)}{\log(1-\beta)} \right\rceil$ uniformly at random from $V(G)$ to find two differently labelled vertices with probability $1 - \alpha$ [Dasarathy et al., 2015]. This means that with probability $1 - \alpha$ the query complexity is bounded by $\text{qc}(G) \leq \left\lceil \frac{\log(\beta\alpha)}{\log(1-\beta)} \right\rceil + \lceil \log d(G) \rceil + \max_{\{a,b\} \in E(G)} |W^*_{=ab}|$. The other alternative is using the mentioned seed queries of Bressan et al. [2021]. Spending only two seed queries to find two vertices with different labels and $\lceil \log d(G) \rceil + \max_{\{a,b\} \in E(G)} |W^*_{=ab}|$ label queries using Algorithm 1 from line 5 on is enough. We note that both bounds can be achieved algorithmically in polynomial time.

**Non-active learning of convex sets and halfspaces**    Non-active learning of halfspaces on graphs was studied by Seiffarth et al. [2019, 2020] and de Araújo et al. [2019]. Seiffarth et al. [2019] rely on the separation axiom $S_4$ to bound the number of convex hull evaluations needed to construct a halfspace. Anthony and Ratsaby [2018] study the problem of learning large-margin halfspaces on a set equipped with an arbitrary *dissimilarity measure* $d$, which generalises shortest path distances of graphs. However, their notion of halfspace, $\{x \mid d(x, p^+) \leq d(x, p^-)\}$ based on the dissimilarity to two prototypes $p^+, p^-$ is in general not convex. Stadtländer et al. [2021] investigate the problem of learning *weakly convex* sets in a metric space $(X, d)$. Here, weakly convex sets with parameter $\theta \geq 0$ are given by the interval mapping $I_\theta(a, b) = \{x \in X \mid d(a, x) + d(x, b) = d(a, b) \leq \theta\} \cup \{a, b\}$. The regular geodesic convexity corresponds to $\theta \geq \max_{a, b \in X} d(a, b)$. Gärtner and Garriga [2007] and Missura and Gärtner [2011] studied monotone classes in directed acyclic graphs, which corresponds to an interval mapping containing the vertices on all directed paths instead of just shortest paths. Moran and Yehudayoff [2019] established that the VC dimension of halfspaces of any convexity space is smaller than the Radon number of the space. Building on that, we can show that VC dimension is exactly determined by the Radon number in $S_4$ convexity spaces. This corresponds to the classical result that in $\mathbb{R}^m$ the VC dimension of halfspaces is $m + 1$ and the Radon number is $m + 2$.

**Proposition 13.** *The VC dimension of the hypothesis class of halfspaces of an $S_4$ convexity space is exactly one less than the Radon number of the space.*

For $S_4$ graphs, $r(G)$ determines the VC dimension and gives a lower bound on the query complexity (Theorem 10), establishing the Radon number as a central parameter for learning geodesically convex halfspaces. Proposition 12 together with the above bound of Moran and Yehudayoff [2019] gives:

**Proposition 14.** *The VC dimension of halfspaces in a weighted graph $G$ is at most $2\,\text{tw}(G) + 1$.*

## 5    Experiments

Having discussed our bounds on the query complexity and the conceptual benefits of the halfspace assumption in terms of theoretical upper bounds, we want to see whether we also get empirical benefits on data, as well. We propose two practical versions of Algorithm 1: greedy and selective sampling. The greedy strategy tries to maximise the number of known vertex labels with each query. For that, it queries the vertex $v$ that would maximise the minimum number of known labels after the update of $A$ and $B$. More precisely, after the update with a vertex $v$ we will know the labels

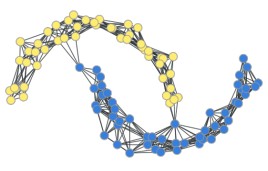
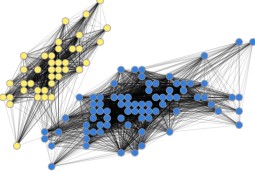
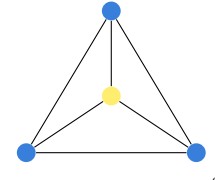

(a) Two moons NN-graph.     (b) Iris dataset NN-graph.     (c) Not separable in $\mathbb{R}^2$.

Figure 1: Halfspace separations on different graphs.

of the set $\sigma((A \cup \{v\})/B) \cup \sigma(B/(A \cup \{v\}))$ or the set $\sigma(A/(B \cup \{v\})) \cup \sigma((B \cup \{v\})/A)$. The `greedy` strategy thus selects the vertex $v$ that maximises the size of the smaller one of these two sets. As performing the described greedy maximisation for each query is rather computationally expensive, we propose a simpler strategy based on `selective sampling` [Cohn et al., 1994] that picks a vertex uniformly at random from $V(G) \setminus (A \cup B)$ in line 8 of Algorithm 1.

To find the first cut edge, Algorithm 1 requires a hull set in the first step. As discussed, computing a minimum hull set is in general not tractable. Therefore, we propose to use the selection strategy (`greedy` or `selective sampling`) also to iteratively construct a hull set until two vertices with different labels are found. For that we start with $A = \emptyset$ and query a vertex $v$ from $V(G) \setminus A$: either uniformly at random (`selective sampling`) or by maximising $|\sigma(A \cup \{v\})|$ (`greedy`). Afterwards we update $A := \sigma(A \cup \{v\})$ and repeat until $v$ has a different label than the previous vertices. Implicitly this process will construct a (typically non-minimum) hull set $H$. That way the query complexity bound in Theorem 5 still holds, as it is independent of the specific selection strategy used; only the $h(G)$ is replaced with the size of the heuristically computed $H$. We compare these two strategies with the state-of-the-art graph-based active learning algorithm $S^2$ [Dasarathy et al., 2015], the classical active label propagation of Zhu et al. [2003b], and baseline non-active random sampling.

As datasets, we use $\varepsilon$-nearest-neighbour graphs of two moons[1] and Iris[2], see Figures 1a and 1b. The parameter $\varepsilon$ was selected such that the graph is connected and the labels are halfspace separable. Additionally, we use a $20 \times 20$ grid and a $2^{10}$ hypercube labelled with a random halfspace. We implemented all approaches in `python`[3] and ran the experiments on an Ubuntu 21.04 laptop with 32GB main memory. More details and in-depth experiments are in the supplementary material.

### 5.1 Query evaluation

Dasarathy et al. [2015] proposed to count the number of found cut vertices after each query as a measure to compare querying algorithms without relying on any classification method. A query-efficient graph-based active learner should detect these as fast as possible. The results for 20 iterative queries in the upper half of Figure 2 show that our approaches identify the cut vertices more efficiently than $S^2$, active label propagation, and random sampling by deducing many labels after each query. Our `greedy` approach requires at most 6 queries to deduce all vertex labels on all four datasets emphasising the strength of convexity-based assumptions. The `selective sampling` approach is only slightly worse using 2-8 queries more. $S^2$, active label propagation, and random sampling can find at most one cut vertex per query as they do not rely on these assumptions. As can be seen on the grid dataset, $S^2$ performs better than active label propagation and random sampling finding a cut vertex on any new query, which is not by coincidence as it was specifically designed for this task.

### 5.2 Predictive performance

We also evaluated the predictive performance of the chosen queries. Our two approaches predict the labels of the computed hulls and default to the majority of known labels when they do not know vertex's label. The other three approaches perform label propagation [Zhu et al., 2003a] with the default Gaussian similarity and length-scale $\sigma = 1$ for prediction. We check the accuracy on the whole graph after each query of the 20 queries. In the lower half of Figure 2, we can confirm that

---

[1] $\varepsilon = 0.14$, `make_moons(noise=0.1, random_state=0)` in `scikit-learn` [Pedregosa et al., 2011]

[2] $\varepsilon = 0.3$, first class vs the other two [Fisher, 1936]

[3] https://github.com/maxthiessen/active_graph_halfspaces

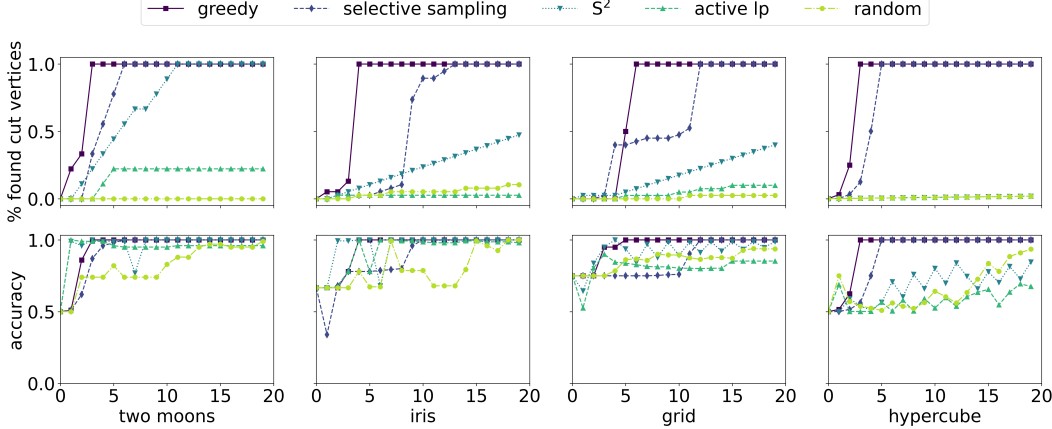

Figure 2: Found cut vertices and accuracy plotted against number of queries.

`greedy` identifies the vertex labels of the whole graph on all datasets within 6 queries. The predictive behaviour of the three baselines is worse, even unstable. The main reason is that label propagation favours small cuts (weighted by the similarity), and thus fails to predict halfspaces, especially on the grid and hypercube. $S^2$ mitigates this problems to some extent, even though it predicts the labels also with label propagation, by removing any found cut edge from the graph. Still, it has difficulties to find the correct halfspace and then stick to these prediction, as can be seen, for example, on the grid.

## 6  Conclusion and Future Work

In this paper, we introduced active learning of halfspaces on graphs and derived tight bounds on the query complexity for this problem using concepts from convexity theory. On the one hand, we derived two general upper bounds: one based on shortest path covers and the other based on the diameter, hull sets, and the quantity $\max_{\{a,b\}\in E(G)} |W^*_{=ab}|$. We showed that the latter term is at most twice the treewidth of the graph in the unweighted case. On the other hand, we derived a general lower bound based on extreme vertices and increasingly tighter lower bounds for graph families induced by the separation axioms. For the strongest separation axiom $S_4$, we achieved nearly tight lower and upper bounds with only the gap between the Radon number $r(G)$ and $\max_{\{a,b\}\in E(G)} |W^*_{=ab}|$ remaining.

We compared our bounds to previous cut-based bounds and showed that the halfspace assumption makes learning with large cuts possible while previous bounds are vacuous in this case. We discussed that there are inherent differences between Euclidean and graph convexity spaces such that the respective query complexity bounds do not carry over to the other setting. With preliminary experiments we confirmed that communities in real-world networks such as collaboration and product networks are often convex. Based on our algorithm, we proposed two practical variants `greedy` and `selective-sampling`. We empirically compared them with active label propagation [Zhu et al., 2003b] and $S^2$ [Dasarathy et al., 2015] on datasets with halfspace separable classes and found that our algorithms require considerably less queries to correctly identify the halfspaces. The main bottleneck of our variants is the cubic runtime to compute convex hulls. To scale our methods to large datasets we will investigate efficient approximations of convex hulls [Blum et al., 2019] and distances using the Nyström [Williams and Seeger, 2001] or other landmark-based methods [Potamias et al., 2009].

To make our results more broadly applicable, we propose to relax our notion of halfspaces and use the weak convexity of Stadtländer et al. [2021]. They define weak convexity with respect to a parameter $\theta \geq 0$ and show that weakly convex sets can be decomposed into weakly convex *blocks* that are at least a distance of $\theta$ apart. The number of these blocks may serve as a parameter to adapt our bounds to weakly convex classes. Another promising future research direction is to develop querying and prediction strategies for general interval convexity spaces by combining our ideas with those of Seiffarth et al. [2019], Stadtländer et al. [2021], and Bressan et al. [2021]. A promising first step is to consider *geometric* interval spaces using *base-point orders* [van de Vel, 1993]. Finally, we propose to also investigate regret bounds for online learning of convex sets on graphs.

**Broader impact statement** As this is an early theoretical and algorithmic work, we do not see any specific immediate societal impact, neither positive nor negative. In the long run the developed theoretical results might help to reduce the required number of labelled data points in graph-based learning settings and help with data annotation in general.

## Acknowledgments and Disclosure of Funding

We thank the anonymous reviewers for helpful comments and Marco Bressan for fruitful discussions.

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
