# Supplementary: Active Learning of Convex Halfspaces on Graphs

**Maximilian Thiessen**
Research Unit of Machine Learning
TU Wien, Vienna, Austria
`maximilian.thiessen@tuwien.ac.at`

**Thomas Gärtner**
Research Unit of Machine Learning
TU Wien, Vienna, Austria
`thomas.gaertner@tuwien.ac.at`

## 1   Proof details

We state some facts that we will often use in the following proofs. For that let $V_+ = \{x \in V(G) \mid \lambda(x) = 1\}$ and $V_- = \{x \in V(G) \mid \lambda(x) = 0\}$ be the labelled halfspaces.

**Fact.**

1. *If $S_+ \subseteq V_+$ then $I(S_+) \subseteq \sigma(S_+) \subseteq V_+$. Equivalently:*
   *If $S_- \subseteq V_-$ then $I(S_-) \subseteq \sigma(S_-) \subseteq V_-$.*

2. *If $S_+ \subseteq V_+$ and $S_- \subseteq V_-$ then $S_+/S_- \subseteq V_+$ and $S_-/S_+ \subseteq V_-$.*

*Proof.*

1. Since $S_+ \subseteq V_+$, we have $\sigma(S_+) \subseteq \sigma(V_+) = V_+$ by the monotonicity of $\sigma(\cdot)$. $I(S_+) \subseteq \sigma(S_+)$ by definition. $S_- \subseteq V_-$ follows analogously.

2. Assume there is an $x \in (S_+/S_-) \cap V_-$. By definition of $S_+/S_-$ there is a $y \in S_+ \cap \sigma(S_- \cup \{x\})$. Using the previous Fact 1, we get $\sigma(S_- \cup \{x\}) \subseteq V_-$ but also $S_+ \subseteq V_+$, which implies that $y \in V_+ \cap V_-$, clearly a contradiction. $S_-/S_+ \subseteq V_-$ follows analogously.

$\square$

**Proposition 1.** *For any weighted graph $G$ with minimum shortest path cover $\mathcal{S}^*$ the query complexity can be bounded as $\mathrm{qc}(G) \leq |\mathcal{S}^*|(2 + \lceil \log d(G) \rceil)$.*

*Proof.* For each path in $\mathcal{S}^*$ first query the endpoints. If they are the same, the whole path must have this label by the halfspace assumption. Otherwise, it has exactly one cut edge, that is, an edge where the labels change. We can find this edge with binary search using at most $\lceil \log d(G) \rceil$ queries. $\square$

**Proposition 2.** *For any $\ell, s \in \mathbb{N}$, there exists a weighted graph $G$ with diameter $d(G) = \ell$ and minimum shortest path cover $\mathcal{S}^*$ of size $s$ such that $\mathrm{qc}(G) \geq |\mathcal{S}^*| \log d(G)$.*

*Proof.* Fix $d, s \in \mathbb{N}$. Consider the unit vectors $e_1, \ldots, e_s \in \mathbb{R}^{s+1}$ and shifted copies of them: $e_j + i \cdot e_{s+1}$, for $j \in \{1, \ldots, s\}$ and $i \in \{1, \ldots, d\}$. Construct the fully connected graph of these $s \cdot d$ points with edge weights corresponding to the Euclidean distance. The minimum shortest path cover will have size $s$ and the diameter will be $d$. We can label each shortest path $P_j$ of the form $(e_j, e_j + e_{s+1}, \ldots, e_j + d \cdot e_{s+1})$ with at most one cut edge at an arbitrary position, independently of the other paths. The resulting labelling is halfspace separable, because the only shortest path between two vertices on different paths $P_j, P_{j'}$ is the edge connecting them. Hence, to deduce all labels we will need at least $\log d(G)$ queries per path. $\square$

35th Conference on Neural Information Processing Systems (NeurIPS 2021).

---

**Algorithm 1:** Halfspace querying

**Input:** graph $G$, oracle access to labels $\lambda$
**Output:** halfspaces corresponding to both classes

**1** Compute a hull set $H$ and query its vertices
**2 if** $\forall h, h' \in H : \lambda(h) = \lambda(h')$ **then**
**3** $\quad$ **return** $(\emptyset, V(G))$
**4** Choose $h, h' \in H$ such that $\lambda(h) \neq \lambda(h')$
**5** Perform binary search queries on any shortest $h$-$h'$-path to find a cut edge $\{a, b\}$
**6** $A := \sigma(a/b), \quad B := \sigma(b/a)$
**7 while** $A \cup B \neq V(G)$ **do**
**8** $\quad$ query any vertex $v \in V(G) \setminus (A \cup B)$
**9** $\quad$ **if** $\lambda(v) = \lambda(a)$ **then**
**10** $\quad\quad$ $A := \sigma((A \cup \{v\})/B), \quad B := \sigma(B/(A \cup \{v\}))$
**11** $\quad$ **else**
**12** $\quad\quad$ $A := \sigma(A/(B \cup \{v\})), \quad B := \sigma((B \cup \{v\})/A)$
**13 return** $(A, B)$

---

**Lemma 3.** *Let $G$ be a weighted graph with halfspace separable labels. Using $h(G) + \lceil \log d(G) \rceil$ queries, we can either find a cut edge or determine that all vertices of the graph have the same label.*

*Proof.* Let $H$ be a hull set of size $h(G)$. The convex hull of $H$ is the whole graph. If all vertices in $H$ have the same label, the whole graph must have this label by Fact 1. Otherwise, we can take two vertices in $H$ with different labels and find one cut edge on a shortest path between them using $\lceil \log d(G) \rceil$ queries. $\qquad\square$

**Lemma 4.** *Let $G$ be a weighted graph with halfspace separable labels given by a halfspace $C$ and let $A \subseteq C$ and $B \subseteq V(G) \setminus C$. It holds that $\sigma(A/B) \subseteq C$.*

*Proof.* The claim directly follows by applying Fact 2 to $A/B$ and then Fact 1 to $\sigma(A/B)$. $\qquad\square$

**Theorem 5.** *Let $G$ be a weighted graph. For the query complexity it holds that:*

$$\mathrm{qc}(G) \leq h(G) + \lceil \log d(G) \rceil + \max_{\{a,b\} \in E(G)} |W^*_{=ab}| \ .$$

*Proof.* By Lemma 3, we can use $h(G) + \lceil \log d(G) \rceil$ queries to find the first cut edge $\{a, b\}$ or deduce that the all vertices have the same label. Applying Lemma 4 to $a/b$ and $b/a$, we only have to deduce the labels of $\hat{W}_{=ab} = V(G) \setminus (\sigma(a/b) \cup \sigma(b/a))$.

Algorithm 1 queries iteratively a vertex label in $V \setminus (\sigma(A/B) \cup \sigma(B/A)) \subseteq \hat{W}_{=ab}$. We want to bound the number of queries made by Algorithm 1 by the size of the set

$$W^*_{=ab} = \{v \in \hat{W}_{=ab} \mid \nexists w \in \hat{W}_{=ab} \setminus \{v\} \text{ such that } v \in (w/a) \cap (w/b)\} \ .$$

For that we show that the algorithm deduces at least one new vertex label from $W^*_{=ab}$ after each update of $A$ and $B$. If we query a vertex in $W^*_{=ab}$ the claim holds, as we never query the same vertex twice, because it is afterwards either in $A$ or $B$.

If we query a vertex $v \in \hat{W}_{=ab} \setminus W^*_{=ab}$, we know by the definition of $W^*_{=ab}$ that there is a vertex $w \in W^*_{=ab}$ with $v \in (w/a) \cap (w/b)$, which means that $w \in \sigma(\{a, v\}) \cap \sigma(\{b, v\})$. By applying Fact 1 to one of the convex hulls $\sigma(\{a, v\})$ or $\sigma(\{b, v\})$, depending on the label of $v$, we get the label of $w$. The vertex $w$ will be hence correctly added to $A$ or $B$ in the following update where the convex hulls $\sigma((A \cup \{v\})/B) \supseteq \sigma(\{a, v\}) \ni w$ or $\sigma((B \cup \{v\})/A) \supseteq \sigma(\{b, v\}) \ni w$ are used.

This shows that after at most $|W^*_{=ab}|$ iterations, we will know the labels of $W^*_{=ab}$. It remains to show that we will also have all missing labels in $\hat{W}_{=ab} \setminus W^*_{=ab}$. For that, let $v \in \hat{W}_{=ab} \setminus W^*_{=ab}$, which implies that there is an already queried vertex $w \in W^*_{=ab}$ such that $v \in (w/a) \cap (w/b)$. Hence, $v$ is correctly added to $A$ or $B$ in the update where $w$ was queried, because $v \in w/a \subseteq \sigma(B \cup \{w\}/A)$ or $v \in w/b \subseteq \sigma(A \cup \{w\}/B)$. $\qquad\square$

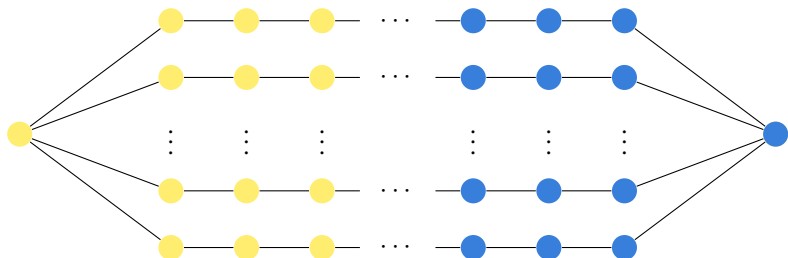

Figure 1: Example graph $P_{s,\ell}$ that is not $S_2$ and has at most two proper halfspaces.

**Corollary 6.** *Let $G$ be an unweighted graph with treewidth* $\mathrm{tw}(G)$*. For the query complexity it holds that* $\mathrm{qc}(G) \leq h(G) + \lceil \log d(G) \rceil + 2\,\mathrm{tw}(G)$.

*Proof.* Let $\{a, b\}$ be a cut edge. We claim that in an unweighted graph, the vertices $a$ and $b$ together with $W^*_{=ab}$ form a $K_{2,|W^*_{=ab}|}$ minor of $G$. To show that, we first prove that any vertex $v \in W^*_{=ab}$ is adjacent to $a/b$ and $b/a$, that is, there is an edge going from $v$ to $a/b$ and to $b/a$. Assume this does not hold. Then, without loss of generality, $v$ is not adjacent to $a/b$ and there exists a $w \in W^*_{=ab}$ such that $w \in I(a, v)$. By the definition of $\hat{W}_{=ab} \supseteq W^*_{=ab}$, the distance from $a$ and $b$ to any vertex in $\hat{W}_{=ab}$ is the same and hence $d(b, v) = d(a, v) = d(a, w) + d(v, w) = d(b, w) + d(v, w)$. This means $w \in I(b, v)$, which implies that $v \in (w/a) \cap (w/b)$. This is a contradiction to $v \in W^*_{=ab}$. It follows, that $v$ is adjacent to $a/b$ and $b/a$.

For each $v \in W^*_{=ab}$, we can identify an edge $e_v = \{v_a, v\}$ with $v_a \in a/b$ on a fixed shortest path from $a$ to $v$. The same holds for a fixed shortest path from $b$ to $v$ with an edge $e'_v = \{v_b, v\}$ and $v_b \in b/a$. Contracting all edges on these shortest paths, but the $e_v$ and $e'_v$ for all $v$ results in a complete bipartite graph $K_{2,|W^*_{=ab}|}$, with the two vertices $a$ and $b$ as one partition and all $v$ as the other. This complete bipartite graph is a minor of $G$. The largest $k$ such that $K_{2,k}$ is a minor of $G$ is bounded as $k \leq 2\,\mathrm{tw}(G)$ by the treewidth [Bodlaender et al., 1997], which in turn bounds $|W^*_{=ab}| \leq k \leq 2\,\mathrm{tw}(G)$. □

**Theorem 7.** *For a weighted graph $G$ with extreme vertices* $\mathrm{Ext}(G)$*, it holds that* $\mathrm{qc}(G) \geq |\,\mathrm{Ext}(G)|$.

*Proof.* As any extreme vertex $s$ is a halfspace, we have $s \notin \sigma(V(G) \setminus \{s\})$. Thus, to decide between the labelling according to the halfspaces $(\{s\}, V(G) \setminus \{s\})$ and the case where all vertices have the same label, we have to query $s$. □

The above result still holds, if we only allow proper halfspaces (neither empty nor $V(G)$) in general, as can be seen by the clique graph where each vertex is extreme.

**Proposition 8.** *For any $\ell, s, k, e \in \mathbb{N}$, there exists an unweighted graph $G$ with $d(G) \geq \ell$, minimum shortest path cover $\mathcal{S}^*$ of size at least $s$, $h(G) \geq k$, $|\,\mathrm{Ext}(G)| = e$, and $\mathrm{qc}(G) \leq \max\{2, e\}$.*

*Proof.* Consider the graph $P_{s,\ell}$ consisting of $s$ interior vertex-disjoint paths of length $\ell$ with same endpoints, see Figure 1. It follows that $d(G) \geq \ell$, the minimum shortest path cover $\mathcal{S}^*$ has size at least $s$ and there are no extreme vertices. We claim that if $\ell$ is even, the graph has only two non-proper halfspaces $V(P_{s,\ell})$ and $\emptyset$ and if $\ell$ is odd additionally two halfspaces cutting each path in two halves of same length.

To show that, we identify three of the paths with vertices $(v_1, \ldots, v_{\ell+1}), (w_1, w_2, \ldots, w_\ell, w_{\ell+1})$, and $(x_1, x_2, \ldots, x_\ell, x_{\ell+1})$. Without loss of generality we can assume that there is a cut edge $v_k, v_{k+1}$ with $k < (\ell+1)/2$ and labels $\lambda(v_k) = 0$, $\lambda(v_{k+1}) = 1$. The ray $v_{k+1}/v_k$ implies by Fact 2 that $\lambda(v_{\ell+1}) = 1$, and the opposite ray $v_k/v_{k+1}$ that $\lambda(v_1) = 0$, $\lambda(w_{\ell+1-k}) = 0$ and $\lambda(x_{\ell+1-k}) = 0$. Finally, $v_{\ell+1} \in I(w_{\ell+1-k}, x_{\ell+1-k})$, which is a contradiction. This implies that a cut edge $\{v_k, v_{k+1}\}$ can only be placed at $k = (\ell+1)/2$ for an odd $\ell$.

This also means that $P_{s,\ell}$ is not $S_2$.

To increase the minimum hull set size, without changing the number of extreme vertices and the query complexity, we append $k$ disjoint copies of $K_{2,3}$ by associating a fixed vertex of $K_{2,3}$ with

exactly on of the endpoints of $P_{s,\ell}$. It can be easily checked that $K_{2,3}$ on its own has only the two halfspaces $V(K_{2,3})$ and $\emptyset$, thus as soon as we know the labels of the endpoints of $P_{s,\ell}$ we also get the labels of the $K_{2,3}$. Each $K_{2,3}$ copy has no extreme vertices but any minimum hull set must contain at least one vertex of each of them, thus $h(G) \geq k$.

Finally to increase the number of extreme vertices we can append $e$ new vertices to either endpoints of $P_{s,\ell}$. These new vertices will be extreme and thus $qc(G) \geq |\text{Ext}(G)| = e$. For $\ell$ even we have $qc(G) = e$. □

*Note that, the graph $P_{2,3}$ is just the complete bipartite $K_{2,3}$.*

**Theorem 11.** *For every weighted graph $G$ the following holds for the query complexity $qc(G)$:*

- *if $G$ is $S_2$, then $qc(G) \geq \max\{\log d(G), |\text{Ext}(G)|\}$,*

- *if $G$ is $S_3$, then $qc(G) \geq \max\{\log d(G), h(G)\}$, and*

- *if $G$ is $S_4$, then $qc(G) \geq \max\{\log d(G), h(G), r(G) - 1\}$.*

*Each bound is tight in the respective family and stronger axioms lead to tighter bounds.*

*Proof.*

- Any two different vertices can be separated with a halfspace ($S_2$):
  The lower bound using extreme vertices $\text{Ext}(G)$ holds for all graphs by Theorem 7 and hence also for $S_2$ graphs. We can place a cut edge anywhere on any fixed shortest path with size of the diameter on any $S_2$ graph, which requires $\log d(G)$ queries to be found in the worst-case.

- Any convex set and a vertex not in this set can be separated with a halfspace ($S_3$):
  On $S_3$ graphs with minimum hull set $H^*$, we can separate any vertex $h \in H^*$ from the ramaining hull set. This means that we have to query all vertices in $H^*$ in the worst-case.

- Any two disjoint convex sets can be separated with a halfspace ($S_4$):
  For any graph, there exists a set $R$ of size $r(G) - 1$ that has no Radon partition. In $S_4$ graphs we can thus extend any partition of $R$ to halfspaces and thus have to query each vertex in $R$ in the worst-case.

  Note that the bounds are increasingly stronger: By the definition of extreme vertices $Ext(G) \subseteq H$ for any hull set $H$. Also, there are $S_4$ graphs with $r(G) = k$ and $h(G) = t$ for any $k, t \in \mathbb{N}$ with $k, t \geq 4$. For that start with a clique of size $t - 2$ and connect two new vertices to each vertex in the clique. Thus $h(G) = 2$ and $r(G) = t$. By attaching $k - 2$ leaf vertices to the graph, we can increase the number of extreme vertices such that $h(G) = k$ without changing the Radon number.

  Regarding the tightness of the bounds: For some example graphs that are $S_i$ but not $S_{i+1}$ for $i \in \{1, 2, 3\}$ and an $S_4$ graph, see Figure 2. For any choice of parameters (diameter, minimum hull set size, Radon number) in the bounds of Table 1 there are graphs achieving these in the respective graph family. That means that we have tight bounds for $S_3$ graphs (that are not $S_4$) and $S_4$ graphs. The lower bound of Theorem 11 matches asymptotically the upper bound of Theorem 5. The same holds for $S_2$ graphs that are not $S_3$, which can be seen by the graph family depicted in Figure 3. Each graph in this graph family is $S_2$ (and not $S_3$) and by adjusting the length of the path in the middle and the number of leaves on top, we can freely select the diameter and the number of extreme vertices. Again, the lower bound and the upper bound on the query complexity asymptotically coincide, as $h(G) = \text{Ext}(G) + 2$ and $|W^*_{=ab}| \leq 1$ for all edges $\{a, b\}$.

  □

We shortly discuss the bounds in Table 1. It contains graph families often studied in metric graph theory [Bandelt and Chepoi, 2008] and convexity theory, such as *partial cubes*, *weakly median* and *meshed* graphs. Meshed graphs satisfy the following *triangle condition* [Bandelt and Chepoi, 2008]: for any three vertices $u, v, w$ with $1 = d(v, w) < d(u, v) = d(u, w)$ there exists a vertex $x$ adjacent

Table 1: Tight bounds for specific unweighted graph families

| graph family | sep. axiom | bound on $\mathrm{qc}(G)$ | remark |
|---|---|---|---|
| trees | $S_4$ | $\Theta(\log d(G) + |\operatorname{Ext}(G)|)$ | $\operatorname{Ext}(G)$ are exactly the leaves |
| $K_{2,3}$ minor-free | $S_4$ | $\Theta(\log d(G) + h(G))$ | including outerplanar graphs |
| partial cubes | $S_3$ | $\Theta(\log d(G) + h(G))$ | $\mathcal{O}(\cdot)$ holds for bipartite graphs |
| weakly median | $S_4$ | $\Theta(\log d(G) + h(G) + r(G))$ | $\mathcal{O}(\cdot)$ holds for meshed graphs |

to $v$ and $w$ such that $d(u,v) = d(u,x) + 1$. For example, the triangle conditions holds for *meshed graphs*, including *weakly modular* graphs. $S_4$ weakly modular graphs are exactly the *weakly median* graphs [Chepoi, 1994]. Bipartite graphs and their $S_3$ variant *partial cubes* [Bandelt, 1989] are also a classical topic in graph convexity theory. For bipartite graphs, $S_2 \iff S_3$. Table 1 also contains trees, and $K_{2,3}$ minor-free graphs a superset of *outerplanar graphs*, studied by Seiffarth et al. [2019].

**Trees** The lower bound follows by Theorem 11, because trees are $S_4$ and $h(G) = |\operatorname{Ext}(G)|$. The upper bound follows by Theorem 5 by the fact that for any edge $\{a,b\}$, we have that $\hat{W}_{=ab} = \emptyset$.

$K_{2,3}$ **minor-free graphs** Seiffarth et al. [2019] showed that $K_{2,3}$ minor-free graphs are $S_4$, which gives the lower bound by Theorem 11. The upper bound follows by Corollary 6. In fact, if $k$ is a constant the upper bound also holds for any $K_{2,k}$ minor-free graph.

**Partial cubes** Partial cubes are exactly the $S_2$ (equivalently $S_3$) bipartite graphs, which gives the lower bound by Theorem 11. The upper bound holds by Theorem 5 as in bipartite graphs for any edge $\{a,b\}$, we have that $\hat{W}_{=ab} = \emptyset$. If there would be an $x \in \hat{W}_{=ab}$ it would form a cycle of odd length with $a$ and $b$, which is not possible in bipartite graphs.

**Weakly median graphs** Weakly median graphs are $S_4$ and hence by Theorem 11 we get the lower bound. The upper bound holds for the much broader class of meshed graphs. In fact, we only need the triangle condition to achieve that the number of main loop iterations of Algorithm 1 is $\leq r(G) - 1$.

By the triangle condition, we have that for any $v \in W^*_{=ab}$, there exists a common neighbour $w$ of $a$ and $b$ that is on the shortest path of $a$ to $v$, respectively $b$ to $v$. So if $d(a,v) = d(b,v) > 1$, we would have $v \in w/a \cup w/b$ and thus $v \notin W^*_{=ab}$, which is not possible. So, $v = w$ and adjacent to $a$ and $b$.

$|W^*_{=ab}| \leq r(G)$ does not hold in general and therefore we can not use the query bound of Theorem 5. Instead, we will directly bound the size of the set of vertices $Q \subseteq W^*_{=ab}$ queried by Algorithm 1. We claim that any two different vertices $v, v' \in Q$ are adjacent. If not, there will be a shortest path from $v$ to $v'$ through $a$ and $b$ respectively, because $v$ and $v'$ are adjacent to $a$ and $b$. This implies $v \in a/v' \cap b/v'$ and $v' \in a/v \cap b/v$. Without loss of generality $v$ is queried before $v'$. Therefore, after the update in Algorithm 1 with $v$, $A$ or $B$ contains $v'$. That means, $v'$ will no be queried, a contradiction. Thus, all vertices in $Q$ are adjacent resulting in a clique. Any clique has no Radon partition, because any subset of a clique is convex, and thus $r(G) > |Q|$.

**Theorem 12.** *For a given weighted graph $G$, we can compute a $\mathcal{O}(\log d(G))$-approximation for the minimum shortest path cover $\mathcal{S}^*$ in time $\mathcal{O}(|V|^4)$.*

*Proof.* We first describe a procedure to compute a shortest path with a maximum number of vertices. More generally, we will describe an iterative procedure covering the vertices of the graph. Each round we will compute a shortest path with a maximum number of vertices that have not been covered yet.

We can assume that the graph is directed, because if it is undirected, we can transform the graph into an equivalent directed graph by duplicating each edge $\{x,y\}$ to $(x,y)$ and $(y,x)$ of same weight. For that, we define new weights $w^*$ for each directed edge $e = (a,b)$ in the graph as $w^*(e) = (w(e), 0)$ if $b$ is already covered and $(w(e), -1)$ otherwise. We note that $w(e) > 0$ for all edges $e$ by assumption. Let us define the lexicographic order $\prec$ on the $w^*$ tuples: $(a,b) \prec (c,d)$ if and only if $(a < b)$ or $(a = b$ and $c < d)$. Note that $\prec$ gives a total order on the $w^*$ tuples. The $w^*$ weight of a path is the component-wise sum of its edges $w^*$ weight. This results in the following behaviour Two paths of

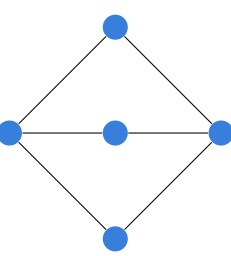

(a) Graph is $S_1$ but not $S_2$. There are no proper halfspaces.

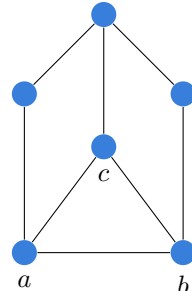

(b) Graph is $S_2$ but not $S_3$. Vertices $a$ and $b$ are not separable from vertex $c$.

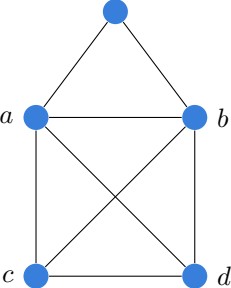

(c) Graph is $S_3$ but not $S_4$. Vertices $a$ and $d$ are not separable from vertices $b$ and $c$.

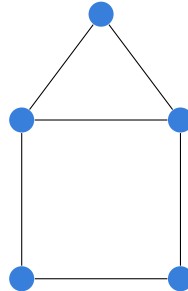

(d) Graph is $S_4$. Any two disjoint convex sets are separable.

Figure 2: Examples for $S_1$ to $S_4$ graphs. Inspired by Bandelt [1989].

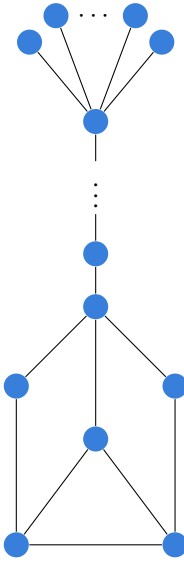

Figure 3: Example of a $S_2$ graph family that is not $S_3$ with tight bounds.

the same $w$-weight, but with one path covering a larger number of not yet covered vertices, have different weights according to $w^*$ and $\prec$. The path covering more vertices has a smaller $w^*$ weight. We thus can apply the generalized Dijkstra's algorithm of Sobrinho [2002] to compute a $w^*$-shortest path, which gives us a $w$-shortest path that covers the maximum number of not yet covered vertices.

Iteratively computing such $w^*$-shortest paths results in a shortest path cover $\mathcal{S}$. Our problem of iteratively covering a graph using shortest path is an instance of the general *set cover problem*. Our covering approach selecting the shortest path covering the maximum number of yet uncovered vertices is an instance of the *greedy* algorithm for the general set cover problem [Chvátal, 1979].

Hence, we achieve at least the same approximation ratio as this more general algorithm, which gives us $|\mathcal{S}| \leq (1 + \ln(d+1))|\mathcal{S}^*|$. $\qquad\square$

**Proposition 13.** *For any graph $G$, it holds that $r(G) \leq 2\operatorname{tw}(G) + 2$.*

*Proof.* Let $\eta(G)$ denote the Hadwiger number of the graph $G$, that is, the size of the largest clique minor. It is known that $\eta(G) \leq \operatorname{tw}(G) + 1$. This can be seen by the fact that any clique $C$ has treewidth $\operatorname{tw}(C) = |C| - 1$ and that any graph minor $F$ of a graph $G$ has treewidth $\operatorname{tw}(F) \leq \operatorname{tw}(G)$, see for example Diestel [2017, Lemma 12.4.1]. Duchet and Meyniel [1983] showed $r(G) \leq 2\eta(G)$. Together this implies $r(G) \leq 2\eta(G) \leq 2\operatorname{tw}(G) + 2$. $\qquad\square$

### 1.1 Passive learning halfspaces

Moran and Yehudayoff [2019] established that the VC dimension of halfspaces of any convexity space is smaller than the Radon number of the space. We prove it here for completeness. Let $(X, \mathcal{C})$ be a convexity space and $r(X, \mathcal{C})$ its Radon number. We only have to show we can not shatter a set $R \subseteq X$ of size $r(X, \mathcal{C})$. By definition, $R$ has a Radon partition $R_1, R_2$ with $\sigma(R_1) \cap \sigma(R_2) \neq \emptyset$. Assume there is a halfspace $C$ separating $R_1$ and $R_2$, that is $R_1 \subseteq C$ and $R_2 \subseteq X \setminus C$. Then by the definition of halfspaces and the monotonicity of the convex hull we have that $\sigma(R_1) \subseteq \sigma(C) = C$ and $\sigma(R_2) \subseteq \sigma(X \setminus C) = X \setminus C$, contradicting that the two sets overlap.

**Proposition 14.** *The VC dimension of the hypothesis class of halfspaces of an $S_4$ convexity space is exactly one less than the Radon number of the space.*

*Proof.* It only remains to show that we can shatter a set $F$ of size $r(G) - 1$ using halfspaces. Let $F$ be a vertex set of size $r(G) - 1$ that has no Radon partition. The convex hulls $\sigma(F_1), \sigma(F_2)$ of any partition $F_1, F_2$ of $F$ do not overlap and hence they can be separated by halfspaces in an $S_4$ space. Thus, halfspaces shatter $F$. $\qquad\square$

**Proposition 15.** *The VC dimension of halfspaces in a graph $G$ is at most $2\operatorname{tw}(G) + 1$.*

*Proof.* Combining the fact of Moran and Yehudayoff [2019] with Proposition 13 it follows that the VC dimension of halfspaces in $G$ is $< r(G) \leq 2\operatorname{tw}(G) + 2$. $\qquad\square$

## 2 Additions to the Euclidean convexity discussion

Graph and other finite convexity spaces are not equivalent to the Euclidean convexity in the sense that it is not always possible to embed a graph into $\mathbb{R}^m$ or the other way around while preserving the convex hull $\sigma(\cdot)$. As Euclidean convexity spaces are always $S_4$ while graphs do not have to be, there is no straightforward way to transform bounds on the query complexity from one setting to the other. But even for $S_4$ interval spaces, the convex hulls given by the interval mapping $I_d$ in a finite metric space $(X, d)$ with $X \subseteq \mathbb{R}^m$ do not always coincide with the classical convex hulls in $(\mathbb{R}^m, d)$.

To see that, take any tree graph with at least 3 leaves, which has the property that the convex hull of any two leaves contains a non-leaf vertex but no additional leaf. However, if we try to reconstruct this behaviour in $\mathbb{R}^m$ for arbitrary $m$, any three points, with at least one non-leaf vertex, must lie on one straight line and hence all points lie on the same straight line. But this implies that either one non-leaf vertex is not contained in the convex hull of one pair of leaves or the convex hull of two leaves contain another leaf.

For the other direction, take the triangle in the Euclidean plane with points $a, b, c$ and a point $d$ in its interior, which has the property that any two points are convex, and the convex hull $\sigma(\{a, b, c\})$ contains $d$, see Fig. 1c) in the main paper. Reconstructing this with any finite metric convexity space, including graphs, is not possible. If we assume convex edges, $I(\{a, b, c\}) = I(a, b) \cup I(a, c) \cup I(b, c) = \{a, b, c\}$, the set $\{a, b, c\}$ is convex and does not contain $d$.

We can phrase this in terms of halfspaces. In the Euclidean plane, $a, b, c$ and $d$ are not halfspace separable, however in any finite metric convexity space that has convex edges they are. In general, if two sets are linearly halfspace separable in Euclidean space, they will be separable by halfspaces in the corresponding fully connected graph with Euclidean distances.

**Margin-based bounds**  We shortly discuss here, why $\mathrm{dens}(V) \geq 1$ for the grid $G = (V, E)$. For a set $A \subseteq V$, let $\mathcal{M}(A, r)$ be the maximum cardinality of any subset $A' \subseteq A$ such that all distinct $a, b \in A'$ satisfy $d(a, b) > r$. Let $B(a, r) = \{b \in V \mid d(a, b) \leq r\}$ denote the ball of radius $r$ around $a$. The density constant of a semimetric space is

$$\mu(V) = \min\{\mu \in \mathbb{N} \mid (v \in V \text{ and } r > 0) \Rightarrow \mathcal{M}(B(v, r), {}^r\!/_2) \leq \mu\}.$$

The density dimension of a semimetric space is then defined as $\mathrm{dens}(V) = \log_2 \mu(V)$.

For an unweighted graph $B(v, 1)$ is simply the set of neighbours of $v$, including $v$. Thus, $\mathcal{M}(B(v, 1), {}^1\!/_2)$ is also again the set of all neighbours $v$, because any pair of vertices $a, b$ has distance $d(a, b) \geq 1 > {}^1\!/_2$. This means that $\mu(V)$ is larger than the maximum degree of the graph. Concluding we have $\mu(V) \geq 2$ for any graph with at least one edge and thus $\mathrm{dens}(V) \geq 1$.

## 3  Experiments

We made all implementations and information on how to reproduce the experiments publicly available[1]. As described in the main text, we evaluated our two approaches *greedy* and *selective sampling* against $S^2$ [Dasarathy et al., 2015], active label propagation [Zhu et al., 2003] and random sampling. We performed 10 independent runs on each dataset, where the first vertex query was the same for all 5 approaches. It was sampled uniformly at random from the same class in the dataset on each run, to guarantee the same starting accuracy for each approach.

For each dataset, the average accuracy after each query is depicted in Figure 4 including error bars representing the $10\%$ and $90\%$ quantile of the 10 runs. The same is visualised in Figure 5 with the average number of found cut vertices after each query instead of the accuracy.

We find that our greedy approach identifies the correct halfspace using 5 to 6 queries most of the time on all datasets. Only occasionally two more queries on the two moons and Iris dataset are required The selective sampling based approach takes some more queries, usually requiring roughly 1.5 to 2 times as many queries as the greedy approach. We thus conclude that, at least on these benchmark datasets the greedy maximisation significantly reduces the number of required queries.

Inspecting the other approaches, we find that they not only take substantially more queries to get close to the correct prediction but also that all of them are unstable. This is meant in the sense that their performance depends a lot on the first drawn vertex and the inherent randomness of the algorithms. Sometimes this even leads to a significant accuracy decrease with more queries, as can be especially be seen on the Iris dataset, see Figure 4b. Here, the accuracy of the predictions start to decrease after the 8th query for $S^2$ and after the 17th query for active label propagation. One reason might be that label propagation favours small cuts (weighted by Gaussian similarity) and hence has problems classifying the cut vertices correctly. Indeed, the Iris dataset has 38 cut edges while two moons only six and here all three label propagation based methods perform well. For example, $S^2$ identifies the correct halfspace after 14 queries.

In terms of the found cut vertices per query, see Figure 5, our approaches use convex hulls and extensions to deduce many labels after each single query. The other approaches do not rely on the halfspace assumptions and hence can only identify at most one cut vertex per query. $S^2$ performs well still, identifying a cut vertex on almost every query after a few initial queries. This is not surprising however, as $S^2$ was designed with explicitly this goal in mind [Dasarathy et al., 2015]. The other two approaches, active label propagation and random sampling, have difficulties finding cut vertices.

## 4  Extensions

**Multi-class settings**  Our results for halfspaces can be generalised to the case where we have $k$ different labels and each corresponding vertex set is convex. Using shortest path covers we get the query complexity upper bound $\mathcal{O}(|\mathcal{S}^*| k \log d(G))$ in this case, as we have to binary search each path at most $k - 1$ times to identify the at most $k - 1$ possible cut edges. Algorithm 1 can be adapted to the multi-class setting but it is not clear how to generalise the $\max\limits_{\{a,b\} \in E(G)} |W^*_{=ab}|$ based upper bound.

---

[1] https://github.com/maxthiessen/active_graph_halfspaces

We are convinced that similar ideas to the ones of Bressan et al. [2021] can be used to achieve an upper bound like $\mathcal{O}\left(h(G) + k^2\left(\log d(G) + \max_{\{a,b\}\in E(G)} |W^*_{=ab}|\right)\right)$.

**Directed graphs**  For directed graphs one has to adjust the definition of the interval mapping $I(x, y)$. The default way [Chartrand and Zhang, 2000, Pelayo, 2013] to fix the symmetry property, $I(x, y) = I(y, x)$, is by first defining a *directed interval* $I_D(x, y) = \{z \mid d(x, z) + d(z, y)\}$, where $d$ is the directed shortest path distance which is not necessarily symmetric, and then simply taking the union of both directions, $I(x, y) \coloneqq I_D(x, y) \cup I_D(y, x)$. The remaining definitions follow as before. Note that this implies that $I(x, y)$ can contain vertices from shortest paths of different length, as the shortest $x$-$y$-path can be shorter than its reversed counterpart.

**Disconnected graphs**  If the graph $G$ is connected all convex vertex sets will induce a connected subgraph. Usually, for a vertex set to be convex in a disconnected graph it is additionally required that the induced subgraph is connected [Pelayo, 2013]. This would correspond to defining $I(x, y) = V(G)$ for any two disconnected vertices $x, y \in V(G)$. Our original query complexity bounds for connected graphs remain valid in this case. If we want to allow halfspace separation in disconnected graphs on multiple connected components, we can ignore this restriction. In this case, we can define the interval $I(x, y) = \{x, y\}$ if $x$ and $y$ are disconnected and keep the original definition $I(x, y) = \{z \in V \mid d(x, y) = d(x, z) + d(z, y)\}$ for connected vertices. Here, we have to query each connected component in isolation, thus the terms of the bounds must be multiplied with the number of connected components $c$, but the minimum hull set size, as it is already the sum of the minimum hull set sizes of each connected component. For example, the bound of Theorem 5 becomes $\mathcal{O}(h(G) + c(\log d(G) + \max_{\{a,b\}\in E(G)} |W^*_{=ab}|))$.

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

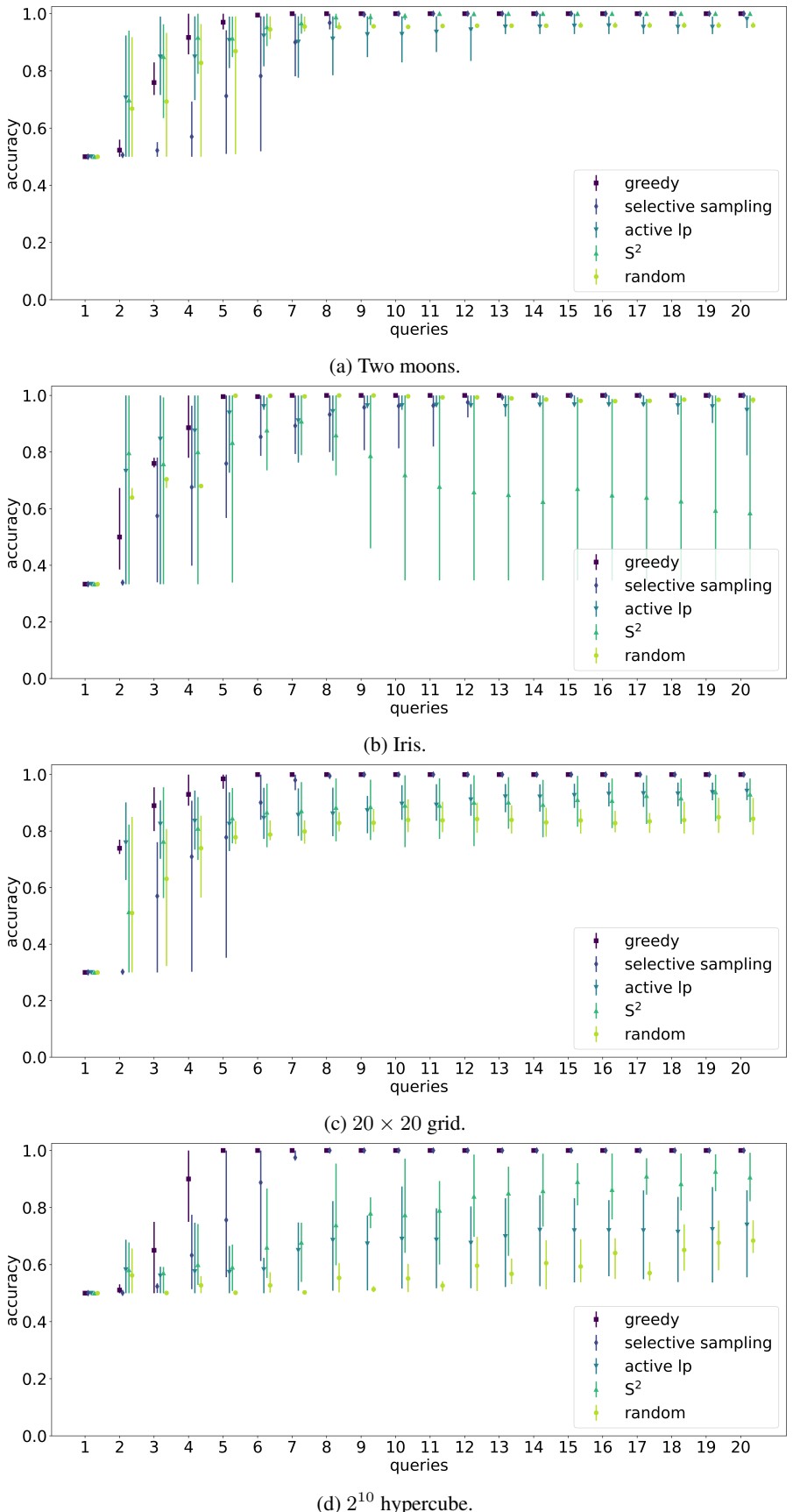

(a) Two moons.

(b) Iris.

(c) $20 \times 20$ grid.

(d) $2^{10}$ hypercube.

Figure 4: Accuracy against number of queries on four benchmark datasets.

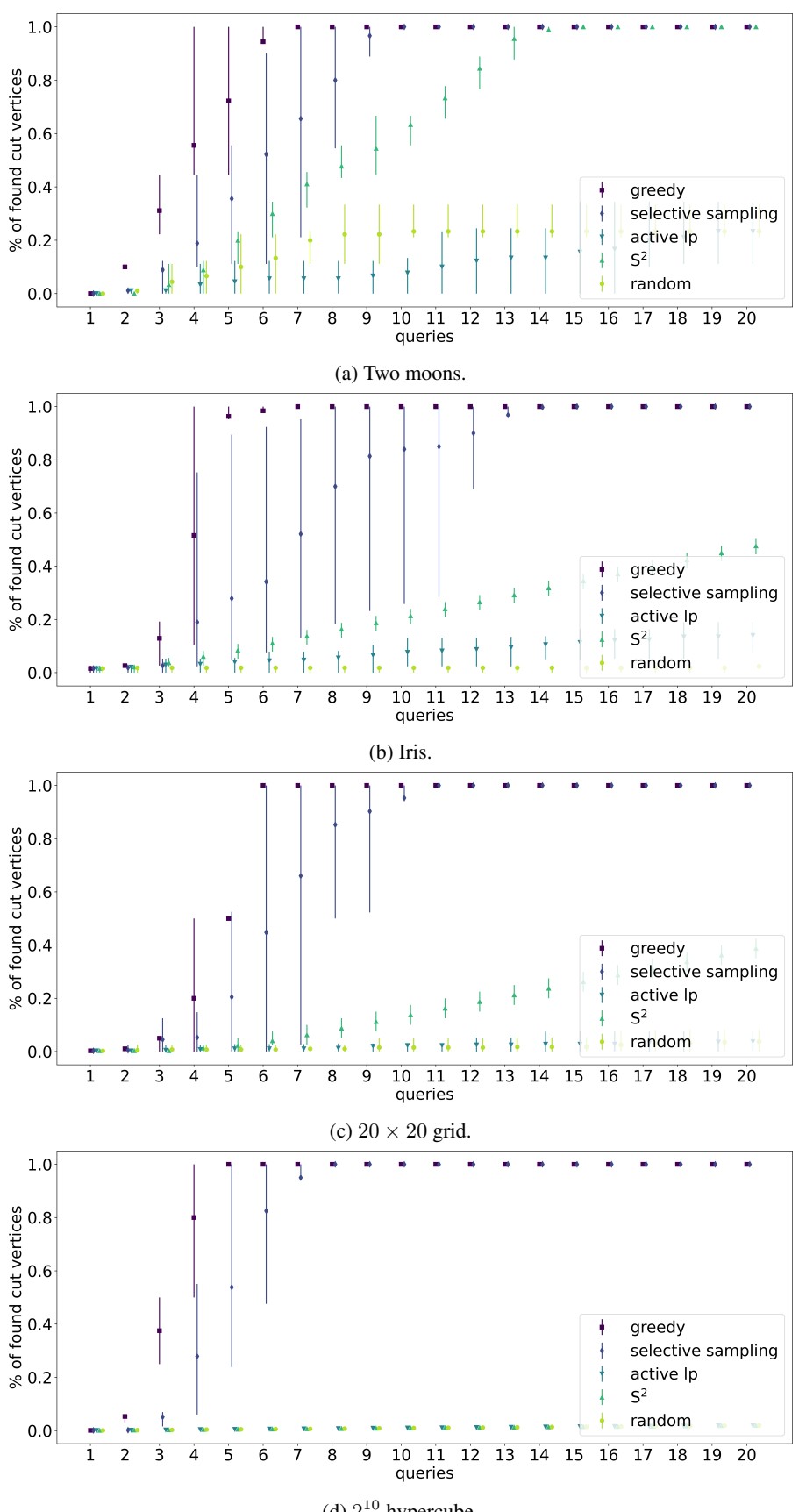

(a) Two moons.

(b) Iris.

(c) $20 \times 20$ grid.

(d) $2^{10}$ hypercube.

Figure 5: Number of found cut vertices against number of queries on four benchmark datasets.