# OpenReview forum: "Active Learning of Convex Halfspaces on Graphs"
_NeurIPS.cc/2021/Conference — NeurIPS 2021 Poster_

### Official Review · Reviewer_KdHb · 2021-07-08

**Rating:** 7
**Confidence:** 3

**Summary:**

This work studies the query complexity of active learning geodesically convex halfspaces on graphs (that is given a graph $G=(V,E)$, the class of subsets $S \subseteq V$ such that $S$ and $V \setminus S$ are both convex). While most prior works gave hypothesis-dependent upper bounds, here the authors give two novel worst-case bounds based on natural graph parameters: the first on the smallest shortest path cover, the second on the hull size, diameter, and tree-width. While neither algorithm is computationally efficient in its original form, the former can be efficiently approximated to within a (multiplicative) logarithmic factor in the diameter, and the latter can be computed efficiently on graphs with bounded tree-width.

The authors also prove a complementary series of lower bounds. The first show both bounds are tight in a weak sense for general graphs in the sense that there exist examples matching the upper bound. They then provide a series of lower bounds based on four (successively stronger) standard convexity properties of the underlying graph $G$. Their final (and tightest) bound holds for graphs on which every two disjoint convex sets are halfspace separable. In this case the only gap between the upper and lower bounds is that between the Radon number and Tree-width of the graph. The authors also provide a number of reasonably natural classes on which their bounds are exactly tight, including trees, $K_{2,3}$ minor-free graphs, partial cubes, and weakly median graphs.

Finally, the authors provide some experimental motivation for their setting, along with a number of heuristic implementations of their algorithm. They compare performance to two non-trivial baseline algorithms, the label propagation algorithm of Zhu et al. (2003), and the $S^2$ algorithm of Dasarathy et al. (2015), and show preliminary experimental evidence that their algorithm outperforms both.

**Limitations And Societal Impact:**

Yes

**Main Review:**

Learning halfspaces is one of the most well-studied problems in machine learning theory. Learning halfspaces on graphs is an understudied variant of this problem, to which this work makes important theoretical contributions in the regime of worst-case analysis. Since learning on graphs is a common problem in practice, this work should be of broad interest both to theoreticians and practitioners. While the techniques themselves used in this work are fairly elementary, the observations are nonetheless novel, and draw interesting connections between standard characterizations of convexity, standard graph complexity parameters, and resulting query complexity lower bounds.

Overall, the paper is well-written and seems technically sound, but could be improved with some discussion of the proofs outside of the appendix (perhaps replacing one of the many discussion sections, or some of the lengthy experimental discussion?). It would also be nice to see more motivation for the study of convex communities in real world graphs, since, while clearly at least related to the topic of study, these don’t seem to obviously lead to halfspace separable learning problems.

A few minor notes for the authors:

1. typos: propsed, conceptional

2. “Same-cluster queries, which are binary queries on a pair of points to test whether they are in the same cluster, instead of vertex label queries” is not a full sentence.

Edit: I thank the authors for their response and my review remains unchanged.

**Time Spent Reviewing:**

8

---

> ### Author Response · Authors · 2021-08-11
> **Response to reviewer KdHb**
>
> Thank you very much for your helpful comments and your interest in our paper.
>
> We will include proof sketches in the revised version using the additional content page and fix the mentioned typos.
>
> ### Regarding convex communities (Table 2)
> As research has only recently started on geodesic convexity assumptions in a learning context (Seiffarth et al. ECMLPKDD 2019, Bressan et al. COLT 2021, Stadtländer et al. ECMLPKDD 2021), we show some large-scale real-world examples to motivate its applicability. Of course, we agree that this does not necessarily lead to halfspace separable classes on graphs. We will improve the motivation for this part of the paper.

---

### Official Review · Reviewer_G7Y8 · 2021-07-16

**Rating:** 6
**Confidence:** 3

**Summary:**

This paper considers learning of geodesically convex halfspaces and present upper and lower bounds on the query complexity.

**Limitations And Societal Impact:**

yes

**Main Review:**

While it seems natural to consider a more general concept class, it is not clear to me the significance of geodesical convexity. What is the main application of the approach?

The technical novelty also appears vague.

Since the paper claims the generality, do the current results recover the best known results of learning halfpsaces in the literature?

If I understand correctly, the results only apply to noise-free learning which is a strong restriction.

**Time Spent Reviewing:**

5

---

> ### Author Response · Authors · 2021-08-11
> **Response to reviewer G7Y8**
>
> Thank you for your helpful comments.
>
> ### Regarding the question on geodesic convexity
> Geodesic halfspaces on graphs, although motivated by halfspace in Euclidean space, are not a generalisation of Euclidean halfspaces. Actually, both are an instance of halfspaces in abstract convexity spaces.
>
> ### Regarding motivations to study halfspaces on graphs
>  * Using the halfspace assumption, we achieve the first label-independent bounds on the query complexity. Previous bounds without the halfspace assumption depend on the labels and can become vacuous in our setting. Label-independent bounds can be computed from the graph itself and thus can be used to get an estimate of the labelling cost before starting the data annotation.
>  * Learning halfspaces in the Euclidean space is a well-known machine learning problem (best paper award NeurIPS 2019 by Diakonikolas et al.). Our graph variant of this problem is promising (cf.reviewer KdHb), as it can lead to a better understanding of learning on graphs in general.
>
> ### Regarding the novelty concern
> We are the first to connect convexity theory and graph-based active learning. Through that, we achieve novel upper and lower bounds on the query complexity. In particular, these are the first label-independent bounds in graph-based active learning. Also, the use of separation axioms is novel in graph-based learning. Note that separation axioms do not provide additional structure in the Euclidean setting, as they all always hold there.
>
> ### Regarding previous bounds for halfspaces
> We are the first to study active halfspace learning on graphs. Previous bounds without the halfspace assumption can become vacuous in our setting (cf. page 6). Additionally, we discuss the relationship to previous bounds in Euclidean space (cf. page 6). There do exist related bounds in the Euclidean setting (e.g., Hopkins et al. 2020), but learning Euclidean halfspaces is not reducible to the graph setting (and vice versa). Hence, bounds from the Euclidean setting do not apply in the graph setting. We will make these point clearer in the revised version.
>
> ### Regarding noise
> Note that sampling techniques can be used to deal with oracle noise. This is discussed in the paper of Dasarathy et al. (2015) and can be applied in our setting. We will clarify this in the revised version and consider different noise conditions in future work.

---

> > ### Comment · Reviewer_G7Y8 · 2021-08-11
> > **reviewer comments**
> >
> > Thank you for the response. Can you give a real-world example where graph-based active learning is useful and will be more favorable than the Euclidean active learning?

---

> > > ### Author Response · Authors · 2021-08-16
> > > **Applications of graph-based active learning**
> > >
> > > A very recent application of graph-based active learning is fault detection in microwave link networks (Regol et al., ICML 2020). The authors state that "the graph is constructed directly from the physical topology and is important because graph-based classification significantly outperforms classification algorithms that ignore the network".
> > >
> > > Another example is protein function prediction on protein-protein-interaction networks, where graph-based methods have been successfully applied (Vazquez et al., Nature Biotechnology 2003). Xiong et al. (Neurocomputing 2014) approached this task using graph-based active learning and improved the TP/FP ratio by 0.1-0.3 compared to random sampling.
> > >
> > > Typically, graph-based active learning is more adequate than Euclidean-based active learning when the data is a graph. For example, Ma et al. (NIPS/NeurIPS 2013) achieved up to 30% better accuracy on graph benchmarks with graph-based active learning compared to uncertainty sampling, which ignores the graph.
> > >
> > > Further applications of graph-based active learning include:
> > >
> > > * Actively searching Wikipedia articles (Garnett et al., KDD 2013)
> > > * Active learning to flatten a hierarchical clustering (Vitale et al., NeurIPS 2019)
> > > * Active learning to make GNNs more label efficient (Hu et al., NeurIPS 2020)

---

> > > > ### Comment · Reviewer_G7Y8 · 2021-08-25
> > > > **thank you**
> > > >
> > > > Thank you for the clarification. I have raised the rating.

---

### Official Review · Reviewer_sgyY · 2021-07-16

**Rating:** 7
**Confidence:** 3

**Summary:**

The paper studies the problem of active learning on graphs to identify two classes. The main assumption is that the vertices of the same class form a geodesically convex set, and the two classes are halfspace separable.
The authors propose a simple halfspace querying algorithm, and analyzes the upper and lower bounds on the number of queries required to deduce all labels of the graph for binary classification on connected undirected graphs. In particular this number of queries is related to the graph properties and the separation condition between two classes. Furthermore the VC dimension of the hypothesis class is related to the aforementioned properties.

**Limitations And Societal Impact:**

n/a
This is a theoretic work.

**Main Review:**

Strength:

The problem discussed in the paper is interesting and novel. If my understanding is correct, the whole assumption at the beginning of Section 3 can be viewed as a generalization of convex and lienar separable clusters in the Euclidean space.

I am not sure how relevent the results are to the NeurIPS community since it is more on the theoretical CS side, but I like the insights about graph properties (treewidth / diameter / hull set) -- upper bound, and separation condition S1, S2, S3, S4 -- lower bound. The latter is in particular interesting, as one does not have the differences in simple spaces.

The paper is well written and organized for most of the time. I did not check the proofs though.



Weakness:

One could potentially argue that the assumption of two classes being geodesically convex and perfectly separable is a very strong condition. Basically this is equivalent to saying that the observed communities are noiseless.

Surprisingly, the paper shows that there exists a large number of convex communities in real world networks as in Table 2. Could the authors comment on how separable (maybe in terms of S1 - S4) the communities are in these real world networks?

**Time Spent Reviewing:**

2

---

> ### Author Response · Authors · 2021-08-11
> **Response to reviewer sgyY**
>
> Thank you very much for your helpful comments and your interest in our work.
>
> We agree that our work is analogous to convex sets and linear separable hypothesis classes in the Euclidean space.
>
> ### Regarding fitting the NeurIPS community
> Learning halfspaces in Euclidean space is a well-known problem in the machine learning community (e.g., best paper award NeurIPS 2019 by Diakonikolas et al.), including many NeurIPS papers studying the active version (e.g., Balcan and Feldman 2013, Hopkins et al. 2020, Zhang et al. 2020). There are many NeurIPS papers dealing with graph-based active learning in the last two decades (e.g., Guillory and Bilmes 2009, Ma et al. 2013, Bressan et al. 2019, Vitale et al. 2019, Hu et al. 2020, Squires et al. 2020).
>
> ### Regarding the noiseless halfspace assumption
> We agree that the (noiseless) halfspace assumption is rather strong. However, even the noiseless variant is valuable (cf. reviewer KdHb), as these are the first label-independent upper bounds in graph-based active learning. It can also lead to a better understanding of learning on graphs in general. Note that, sampling techniques can be used to deal with oracle noise. This is discussed in the paper by Dasarathy et al. (2015) and can be applied in our setting. We will clarify this in the revised version. We will consider different noise conditions and relaxations of our assumptions in future work.
>
> ### Regarding convexity of real-world networks (Table 2)
> As research has only recently started on geodesic convexity assumptions in a learning context (Seiffarth et al. ECMLPKDD 2019, Bressan et al. COLT 2021, Stadtländer et al. ECMLPKDD 2021), we show some large-scale real-world examples to motivate its applicability. We will clarify this in our revised version. Regarding the separability of communities, note that communities in community detection datasets are often overlapping (multilabel). Thus it is not possible to separate them.
>
> ### Regarding separability of real-world networks (Table 2)
> The separation axioms are properties of the graph and do not depend on labels/communities. Thus, we could only check if the given graphs satisfy any of the four separation axioms. However, for axioms S1 to S3 no polynomial-time algorithm is known, and for the S4 axiom only an inefficient one exists (with |V|^7 runtime).

---

### Official Review · Reviewer_mSJZ · 2021-07-18

**Rating:** 7
**Confidence:** 2

**Summary:**

This paper studies the query complexity of learning geodesically convex halfspaces on graphs.  In the present context, a geodesically convex set of a graph $G = (V,E)$ is a subset $C$ of $V$ such that each shortest path containing any two vertices of $C$ is contained in $C$, and a halfspace of $G$ is any convex set of $G$ whose complement is also convex.  The main contributions are upper and lower bounds for the query complexity of a class of halfspaces on a weighted graph in terms of various graph parameters.  One upper and almost matching lower bound is given in terms of the size of the minimum shortest cover and the graph diameter.  Another upper bound is given in terms of the minimum size of a hull set, the graph diameter and the maximum size of a set $W_{=ab}^*$ of vertices over all pairs $(a,b)$ of vertices, where $W_{=ab}^*$ is defined to be the set of all points that avoid the intersection of rays from $a$ (resp. $b$) to any point neither in the convex hull of $a/b$ nor in that of $b/a$ and are themselves not in these convex hulls.  The size of the set $W_{=ab}^*$ can also be bounded in terms of the treewidth.  It is shown, moreover, that increasingly tighter lower bounds can be obtained as more stringent separation axioms are assumed.


**Ethical Concerns:**

None.

**Limitations And Societal Impact:**

Yes, I think so.

**Main Review:**

I checked the proofs in the Supplementary up to and including Theorem 10; I think they are correct.  The remaining proofs also look convincing enough.  Overall, I think the results in the paper are correct.

However, I am not familiar with the area so I cannot assess how novel the proof methods/conceptual ideas are.  It seems from the writing that the paper does not introduce any new proof techniques, so the focus seems to be mainly on the application of known results in metric graph theory and general convex structures to the problem of learning graph labels under a new assumption, namely that each class of labels forms a halfspace.

Comments/suggestions:

- Page 3, Lemma 4 and Supplementary, page 2, Lemma 4: I suggest phrasing Lemma 4 in the main text as well as the Supplementary as follows: Using $h(G) + \lceil log d(G) \rceil$ queries, we can either find a cut edge or determine that all vertices of the graph have the same label.


Minor comments/suggestions:

- Supplementary, page 1, line 19: change -> changes

- Supplementary, page 4, line 103: $d$ -> $d(G)$

Edit: I raised my score to 7 after reading the author(s)' explanations on the novelty of this work.

**Time Spent Reviewing:**

16

---

> ### Author Response · Authors · 2021-08-11
> **Response to reviewer mSJZ**
>
> Thank you very much for the helpful review and for confirming our theoretical results. We will rewrite Lemma 4 according to your suggestion.
>
> ### Regarding novelty:
>
>  * We introduce the problem of active learning halfspaces on graphs and achieve label-independent query complexity bounds. Previous bounds without the halfspace assumption depend on the labels and can become vacuous in our setting.
>  * The connection between learning on graphs and convexity theory is understudied; only recently research has started on this topic (Seiffarth et al. ECMLPKDD 2019, Bressan et al. COLT 2021, Stadtländer et al. ECMLPKDD 2021). The idea of structuring the set of all graphs along separation axioms S1,..,S4 to achieve lower bounds is novel. We emphasise that separation axioms do not provide additional structure in the Euclidean setting, as all separation axioms always hold there.
>  * The identification of the Radon number as a parameter for the VC dimension and query complexity bounds is new in this setting.
>  * We use the treewidth in our query complexity upper bound as a parameter. Treewidth is a well-known parameter in graph theory and we are unaware of any other query (or sample) complexity bound using it.
>  * Our analysis using $W_{a,b}^*$ sets is also novel.

---

> > ### Comment · Reviewer_mSJZ · 2021-08-29
> > **Response to Response to reviewer mSJZ**
> >
> > Thank you very much for the detailed explanations on the novelty of this work.  I agree that the use of separation axioms to classify the query complexity of various weighted graphs is interesting and new.  The observation of convexity in real-world data sets seems to me quite intriguing too, though I suppose this was not the main emphasis of the paper; as the paper observed, convexity seems to be application-dependent (e.g. strong convexity in the DBLP dataset but weak convexity in the LiveJournal dataset) and there does not seem to be much explanation in the paper for the differences in degrees of convexities of different real-world graphs.  Furthermore, as Reviewer sgyY commented, the assumption that the graph is perfectly halfspace separable seems quite strong, though from a theoretical point of view I think it is a good starting point for analysis.  Overall, I think the response to the initial reviews was very good (particularly the elaboration on applications of graph-based active learning), and will raise my score to 7.

---

### Decision · Program_Chairs · 2021-09-28

**Decision:**

Accept (Poster)

**Comment:**

This is an original work which proposes and explores a natural and elegant abstraction of convexity. The reviewers unanimously agreed to accept this paper.

**Consistency Experiment:**

NeurIPS has a long history of experimentation. In 2014, NeurIPS ran an experiment in which 10% of submissions were reviewed by two independent committees to quantify the randomness in the review process. This year, we repeated a variant of this experiment to see how the quality of the review process has changed over time.  This paper was part of the experiment and was therefore assigned to two committees (consisting of reviewers, an Area Chair, and a Senior Area Chair) that reached independent decisions.  If both committees made the same recommendation, this recommendation was followed. If a single committee recommended acceptance, the paper was accepted (with the exception of a few cases in which the other committee identified what we considered a fatal flaw, e.g., an error in a key result).

Both committees reached the same decision: **Accept (Poster)**

The other committee assigned to the paper recommended **Accept (Poster)**.  You can find the other set of reviews, along with any follow up discussion with the authors here:
https://openreview.net/forum?id=O-fOgeI_D-B